# Meta-Weight-Net: Learning an Explicit Mapping For Sample Weighting

Jun Shu[1], Qi Xie[1], Lixuan Yi[1], Qian Zhao[1], Sanping Zhou[1], Zongben Xu[1], and Deyu Meng*[2,1]

[1]Xi'an Jiaotong University
[2]The Macau University of Science and Technology
*Corresponding author:dymeng@mail.xjtu.edu.cn

## Abstract

Current deep neural networks (DNNs) can easily overfit to biased training data with corrupted labels or class imbalance. Sample re-weighting strategy is commonly used to alleviate this issue by designing a weighting function mapping from training loss to sample weight, and then iterating between weight recalculating and classifier updating. Current approaches, however, need manually pre-specify the weighting function as well as its additional hyper-parameters. It makes them fairly hard to be generally applied in practice due to the significant variation of proper weighting schemes relying on the investigated problem and training data. To address this issue, we propose a method capable of adaptively learning an explicit weighting function directly from data. The weighting function is an MLP with one hidden layer, constituting a universal approximator to almost any continuous functions, making the method able to fit a wide range of weighting functions including those assumed in conventional research. Guided by a small amount of unbiased meta-data, the parameters of the weighting function can be finely updated simultaneously with the learning process of the classifiers. Synthetic and real experiments substantiate the capability of our method for achieving proper weighting functions in class imbalance and noisy label cases, fully complying with the common settings in traditional methods, and more complicated scenarios beyond conventional cases. This naturally leads to its better accuracy than other state-of-the-art methods. Source code is available at `https://github.com/xjtushujun/meta-weight-net`.

## 1 Introduction

DNNs have recently obtained impressive good performance on various applications due to their powerful capacity for modeling complex input patterns. However, DNNs can easily overfit to biased training data[1], like those containing corrupted labels [2] or with class imbalance[3], leading to their poor performance in generalization in such cases. This robust deep learning issue has been theoretically illustrated in multiple literatures [4, 5, 6, 7, 8, 9].

In practice, however, such biased training data are commonly encountered. For instance, practically collected training samples always contain corrupted labels [10, 11, 12, 13, 14, 15, 16, 17]. A typical example is a dataset roughly collected from a crowdsourcing system [18] or search engines [19, 20], which would possibly yield a large amount of noisy labels. Another popular type of biased training data is those with class imbalance. Real-world datasets are usually depicted as skewed distributions, with a long-tailed configuration. A few classes account for most of the data, while most classes are

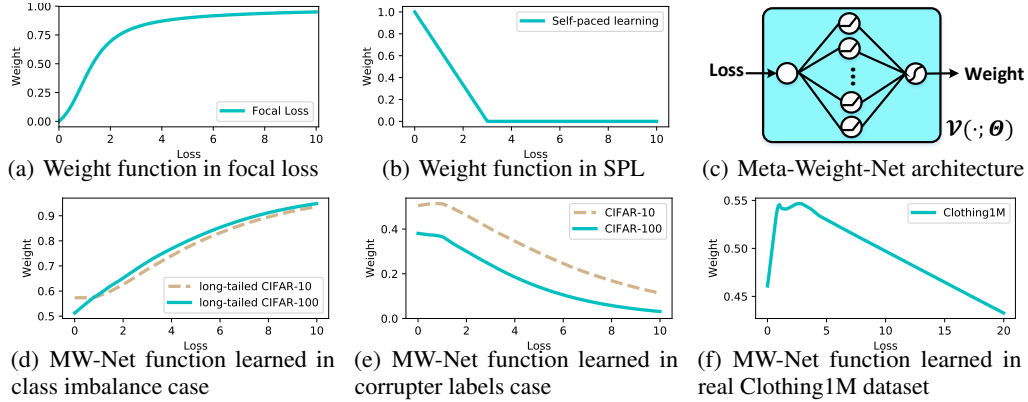

Figure 1: (a)-(b) weight functions set in focal loss and self-paced learning (SPL). (c) Meta-Weighting-Net architecture. (d)-(f) Meta-Weighting-Net functions learned in class imbalance (imbalanced factor 100), noisy label (40% uniform noise), and real dataset, respectively, by our method.

under-represented. Effective learning with these biased training data, which is regarded to be biased from evaluation/test ones, is thus an important while challenging issue in machine learning [1, 21].

Sample reweighting approach is a commonly used strategy against this robust learning issue. The main methodology is to design a weighting function mapping from training loss to sample weight (with hyper-parameters), and then iterates between calculating weights from current training loss values and minimizing weighted training loss for classifier updating. There exist two entirely contradictive ideas for constructing such a loss-weight mapping. One makes the function monotonically increasing as depicted in Fig. 1(a), i.e., enforce the learning to more emphasize samples with larger loss values since they are more like to be uncertain hard samples located on the classification boundary. Typical methods of this category include AdaBoost [22, 23], hard negative mining [24] and focal loss [25]. This sample weighting manner is known to be necessary for class imbalance problems, since it can prioritize the minority class with relatively higher training losses.

On the contrary, the other methodology sets the weighting function as monotonically decreasing, as shown in Fig. 1(b), to take samples with smaller loss values as more important ones. The rationality lies on that these samples are more likely to be high-confident ones with clean labels. Typical methods include self-paced learning(SPL) [26], iterative reweighting [27, 17] and multiple variants [28, 29, 30]. This weighting strategy has been especially used in noisy label cases, since it inclines to suppress the effects of samples with extremely large loss values, possibly with corrupted incorrect labels.

Although these sample reweighting methods help improve the robustness of a learning algorithm on biased training samples, they still have evident deficiencies in practice. On the one hand, current methods need to manually set a specific form of weighting function based on certain assumptions on training data. This, however, tends to be infeasible when we know little knowledge underlying data or the label conditions are too complicated, like the case that the training set is both imbalanced and noisy. On the other hand, even when we specify certain weighting schemes, like focal loss [25] or SPL [26], they inevitably involve hyper-parameters, like focusing parameter in the former and age parameter in the latter, to be manually preset or tuned by cross-validation. This tends to further raise their application difficulty and reduce their performance stability in real problems.

To alleviate the aforementioned issue, this paper presents an adaptive sample weighting strategy to automatically learn an explicit weighting function from data. The main idea is to parameterize the weighting function as an MLP (multilayer perceptron) network with only one hidden layer (as shown in Fig. 1(c)), called ***Meta-Weight-Net***, which is theoretically a universal approximator for almost any continuous function [31], and then use a small unbiased validation set (meta-data) to guide the training of all its parameters. The explicit form of the weighting function can be finally attained specifically suitable to the learning task.

In summary, this paper makes the following three-fold contributions:

1) We propose to automatically learn an explicit loss-weight function, parameterized by an MLP from data in a meta-learning manner. Due to the universal approximation capability of this weight net, it can finely fit a wide range of weighting functions including those used in conventional research.

2) Experiments verify that the weighting functions learned by our method highly comply with manually preset weighting manners used in tradition in different training data biases, like class imbalance and noisy label cases as shown in Fig. 1(d) and 1(e)), respectively. This shows that the weighting scheme learned by the proposed method inclines to help reveal deeper understanding for data bias insights, especially in complicated bias cases where the extracted weighting function is with complex tendencies (as shown in Fig. 1(f)).

3) The insights of why the proposed method works can be well interpreted. Particularly, the updating equation for Meta-Weight-Net parameters can be explained by that the sample weights of those samples better complying with the meta-data knowledge will be improved, while those violating such meta-knowledge will be suppressed. This tallies with our common sense on the problem: we should reduce the influence of those highly biased ones, while emphasize those unbiased ones.

The paper is organized as follows. Section 2 presents the proposed meta-learning method as well as the detailed algorithm and analysis of its convergence property. Section 3 discusses related work. Section 4 demonstrates experimental results and the conclusion is finally made.

## 2   The Proposed Meta-Weight-Net Learning Method

### 2.1   The Meta-learning Objective

Consider a classification problem with the training set $\{x_i, y_i\}_{i=1}^N$, where $x_i$ denotes the $i$-th sample, $y_i \in \{0, 1\}^c$ is the label vector over $c$ classes, and $N$ is the number of the entire training data. $f(x, \mathbf{w})$ denotes the classifier, and $\mathbf{w}$ denotes its parameters. In current applications, $f(x, \mathbf{w})$ is always set as a DNN. We thus also adopt DNN, and call it the classifier network for convenience in the following.

Generally, the optimal classifier parameter $\mathbf{w}^*$ can be extracted by minimizing the loss $\frac{1}{N}\sum_{i=1}^N$ $\ell(y_i, f(x_i, \mathbf{w}))$ calculated on the training set. For notation convenience, we denote that $L_i^{train}(\mathbf{w}) = \ell(y_i, f(x_i, \mathbf{w}))$. In the presence of biased training data, sample re-weighting methods enhance the robustness of training by imposing weight $\mathcal{V}(L_i^{train}(\mathbf{w}); \Theta)$ on the $i$-th sample loss, where $\mathcal{V}(\ell; \Theta)$ denotes the weight net, and $\Theta$ represents the parameters contained in it. The optimal parameter $\mathbf{w}$ is calculated by minimizing the following weighted loss:

$$\mathbf{w}^*(\Theta) = \arg\min_{\mathbf{w}} \ \mathcal{L}^{train}(\mathbf{w}; \Theta) \triangleq \frac{1}{N}\sum_{i=1}^N \mathcal{V}(L_i^{train}(\mathbf{w}); \Theta) L_i^{train}(\mathbf{w}). \qquad (1)$$

**Meta-Weight-Net:** Our method aims to automatically learn the hyper-parameters $\Theta$ in a meta-learning manner. To this aim, we formulate $\mathcal{V}(L_i(\mathbf{w}); \Theta)$ as a MLP network with only one hidden layer containing 100 nodes, as shown in Fig. 1(c). We call this weight net as ***Meta-Weight-Net*** or ***MW-Net*** for easy reference. Each hidden node is with ReLU activation function, and the output is with the Sigmoid activation function, to guarantee the output located in the interval of $[0, 1]$. Albeit simple, this net is known as a universal approximator for almost any continuous function [31], and thus can fit a wide range of weighting functions including those used in conventional research.

**Meta learning process.** The parameters contained in MW-Net can be optimized by using the meta learning idea [32, 33, 34, 35]. Specifically, assume that we have a small amount unbiased meta-data set (i.e., with clean labels and balanced data distribution) $\{x_i^{(meta)}, y_i^{(meta)}\}_{i=1}^M$, representing the meta-knowledge of ground-truth sample-label distribution, where $M$ is the number of meta-samples and $M \ll N$. The optimal parameter $\Theta^*$ can be obtained by minimizing the following meta-loss:

$$\Theta^* = \arg\min_{\Theta} \ \mathcal{L}^{meta}(\mathbf{w}^*(\Theta)) \triangleq \frac{1}{M}\sum_{i=1}^M L_i^{meta}(\mathbf{w}^*(\Theta)), \qquad (2)$$

where $L_i^{meta}(\mathbf{w}) = \ell\left(y_i^{(meta)}, f(x_i^{(meta)}, \mathbf{w})\right)$ is calculated on meta-data.

### 2.2   The Meta-Weight-Net Learning Method

Calculating the optimal $\Theta^*$ and $\mathbf{w}^*$ require two nested loops of optimization. Here we adopt an online strategy to update $\Theta$ and $\mathbf{w}$ through a single optimization loop, respectively, to guarantee the efficiency of the algorithm.

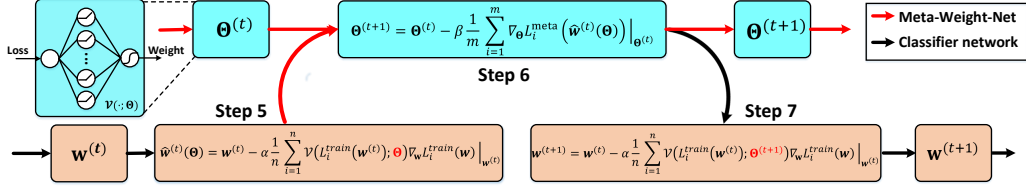

Figure 2: Main flowchart of the proposed MW-Net Learning algorithm (steps 5-7 in Algorithm 1).

**Formulating learning manner of classifier network.** As general network training tricks, we employ SGD to optimize the training loss (1). Specifically, in each iteration of training, a mini-batch of training samples $\{(x_i, y_i), 1 \leq i \leq n\}$ is sampled, where $n$ is the mini-batch size. Then the updating equation of the classifier network parameter can be formulated by moving the current $\mathbf{w}^{(t)}$ along the descent direction of the objective loss in Eq. (1) on a mini-batch training data:

$$\hat{\mathbf{w}}^{(t)}(\Theta) = \mathbf{w}^{(t)} - \alpha \frac{1}{n} \times \sum_{i=1}^{n} \mathcal{V}(L_i^{train}(\mathbf{w}^{(t)}); \Theta) \nabla_{\mathbf{w}} L_i^{train}(\mathbf{w}) \Big|_{\mathbf{w}^{(t)}}, \tag{3}$$

where $\alpha$ is the step size.

---

**Algorithm 1** The MW-Net Learning Algorithm

---

**Input:** Training data $\mathcal{D}$, meta-data set $\widehat{\mathcal{D}}$, batch size $n, m$, max iterations $T$.
**Output:** Classifier network parameter $\mathbf{w}^{(T)}$
1: Initialize classifier network parameter $\mathbf{w}^{(0)}$ and Meta-Weight-Net parameter $\Theta^{(0)}$.
2: **for** $t = 0$ **to** $T - 1$ **do**
3:     $\{x, y\} \leftarrow$ SampleMiniBatch($\mathcal{D}, n$).
4:     $\{x^{(meta)}, y^{(meta)}\} \leftarrow$ SampleMiniBatch($\widehat{\mathcal{D}}, m$).
5:     Formulate the classifier learning function $\hat{\mathbf{w}}^{(t)}(\Theta)$ by Eq. (3).
6:     Update $\Theta^{(t+1)}$ by Eq. (4).
7:     Update $\mathbf{w}^{(t+1)}$ by Eq. (5).
8: **end for**

---

**Updating parameters of Meta-Weight-Net:** After receiving the feedback of the classifier network parameter updating formulation $\hat{\mathbf{w}}^{(t)}(\Theta)$ [2] from the Eq .(3), the parameter $\Theta$ of the Meta-Weight-Net can then be readily updated guided by Eq. (2), i.e., moving the current parameter $\Theta^{(t)}$ along the objective gradient of Eq. (2) calculated on the meta-data:

$$\Theta^{(t+1)} = \Theta^{(t)} - \beta \frac{1}{m} \sum_{i=1}^{m} \nabla_{\Theta} L_i^{meta}(\hat{\mathbf{w}}^{(t)}(\Theta)) \Big|_{\Theta^{(t)}}, \tag{4}$$

where $\beta$ is the step size.

**Updating parameters of classifier network:** Then, the updated $\Theta^{(t+1)}$ is employed to ameliorate the parameter $\mathbf{w}$ of the classifier network, i.e.,

$$\mathbf{w}^{(t+1)} = \mathbf{w}^{(t)} - \alpha \frac{1}{n} \times \sum_{i=1}^{n} \mathcal{V}(L_i^{train}(\mathbf{w}^{(t)}); \Theta^{(t+1)}) \nabla_{\mathbf{w}} L_i^{train}(\mathbf{w}) \Big|_{\mathbf{w}^{(t)}}. \tag{5}$$

The MW-Net Learning algorithm can then be summarized in Algorithm 1, and Fig. 2 illustrates its main implementation process (steps 5-7). All computations of gradients can be efficiently implemented by automatic differentiation techniques and generalized to any deep learning architectures of classifier network. The algorithm can be easily implemented using popular deep learning frameworks like PyTorch [36]. It is easy to see that both the classifier network and the MW-Net gradually ameliorate their parameters during the learning process based on their values calculated in the last step, and the weights can thus be updated in a stable manner, as clearly shown in Fig. 6.

## 2.3 Analysis on the Weighting Scheme of Meta-Weight-Net

The computation of Eq. (4) by backpropagation can be rewritten as[3]:

$$\Theta^{(t+1)} = \Theta^{(t)} + \frac{\alpha\beta}{n} \sum_{j=1}^{n} \left( \frac{1}{m} \sum_{i=1}^{m} G_{ij} \right) \frac{\partial \mathcal{V}(L_j^{train}(\mathbf{w}^{(t)}); \Theta)}{\partial \Theta} \bigg|_{\Theta^{(t)}}, \tag{6}$$

where $G_{ij} = \frac{\partial L_i^{meta}(\hat{\mathbf{w}})}{\partial \hat{\mathbf{w}}}\big|_{\hat{\mathbf{w}}^{(t)}}^{T} \frac{\partial L_j^{train}(\mathbf{w})}{\partial \mathbf{w}}\big|_{\mathbf{w}^{(t)}}$. Neglecting the coefficient $\frac{1}{m}\sum_{i=1}^{m} G_{ij}$, it is easy to see that each term in the sum orients to the ascend gradient of the weight function $\mathcal{V}(L_j^{train}(\mathbf{w}^{(t)}); \Theta)$. $\frac{1}{m}\sum_{i=1}^{m} G_{ij}$, the coefficient imposed on the $j$-th gradient term, represents the similarity between the gradient of the $j$-th training sample computed on training loss and the average gradient of the mini-batch meta data calculated on meta loss. That means if the learning gradient of a training sample is similar to that of the meta samples, then it will be considered as beneficial for getting right results and its weight tends to be more possibly increased. Conversely, the weight of the sample inclines to be suppressed. This understanding is consistent with why well-known MAML works [37, 38, 39].

## 2.4 Convergence of the MW-Net Learning algorithm

Our algorithm involves optimization of two-level objectives, and therefore we show theoretically that our method converges to the critical points of both the meta and training loss function under some mild conditions in Theorem 1 and 2, respectively. The proof is listed in the supplementary material.

**Theorem 1.** *Suppose the loss function $\ell$ is Lipschitz smooth with constant L, and $\mathcal{V}(\cdot)$ is differential with a $\delta$-bounded gradient and twice differential with its Hessian bounded by $\mathcal{B}$, and the loss function $\ell$ have $\rho$-bounded gradients with respect to training/meta data. Let the learning rate $\alpha_t$ satisfies $\alpha_t = \min\{1, \frac{k}{T}\}$, for some $k > 0$, such that $\frac{k}{T} < 1$, and $\beta_t, 1 \leq t \leq N$ is a monotone descent sequence, $\beta_t = \min\{\frac{1}{L}, \frac{c}{\sigma\sqrt{T}}\}$ for some $c > 0$, such that $\frac{\sigma\sqrt{T}}{c} \geq L$ and $\sum_{t=1}^{\infty} \beta_t \leq \infty, \sum_{t=1}^{\infty} \beta_t^2 \leq \infty$. Then the proposed algorithm can achieve $\mathbb{E}[\|\nabla G(\Theta^{(t)})\|_2^2] \leq \epsilon$ in $\mathcal{O}(1/\epsilon^2)$ steps. More specifically,*

$$\min_{0 \leq t \leq T} \mathbb{E}[\|\nabla \mathcal{L}^{meta}(\Theta^{(t)})\|_2^2] \leq \mathcal{O}(\frac{C}{\sqrt{T}}), \tag{7}$$

*where $C$ is some constant independent of the convergence process, and $\sigma$ is the variance of drawing uniformly mini-batch sample at random.*

**Theorem 2.** *The condions in Theorem 1 hold, then we have:*

$$\lim_{t \to \infty} \mathbb{E}[\|\nabla \mathcal{L}^{train}(\mathbf{w}^{(t)}; \Theta^{(t+1)})\|_2^2] = 0. \tag{8}$$

# 3 Related Work

**Sample Weighting Methods.** The idea of reweighting examples can be dated back to dataset resampling [40, 41] or instance re-weight [42], which pre-evaluates the sample weights as a pre-processing step by using certain prior knowledge on the task or data. To make the sample weights fit data more flexibly, more recent researchers focused on pre-designing a weighting function mapping from training loss to sample weight, and dynamically ameliorate weights during training process [43, 44]. There are mainly two manners to design the weighting function. One is to make it monotonically increasing, specifically effective in class imbalance case. Typical methods include the boosting algorithm (like AdaBoost [22]) and multiple of its variations [45], hard example mining [24] and focal loss [25], which impose larger weights to ones with larger loss values. On the contrary, another series of methods specify the weighting function as monotonically decreasing, especially used in noisy label cases. For example, SPL [26] and its extensions [28, 29], iterative reweighting [27, 17] and other recent work [46, 30], pay more focus on easy samples with smaller losses. The limitation of these methods are that they all need to manually pre-specify the form of weighting function as well as their hyper-parameters, raising their difficulty to be readily used in real applications.

**Meta Learning Methods.** Inspired by meta-learning developments [47, 48, 49, 37, 50], recently some methods were proposed to learn an adaptive weighting scheme from data to make the learning more automatic and reliable. Typical methods along this line include FWL [51], learning to teach [52, 32] and MentorNet [21] methods, whose weight functions are designed as a Bayesian function approximator, a DNN with attention mechanism, a bidirectional LSTM network, respectively. Instead of only taking loss values as inputs as classical methods, the weighting functions they used (i.e., the meta-learner), however, are with much more complex forms and required to input complicated information (like sample features). This makes them not only hard to succeed good properties possessed by traditional methods, but also to be easily reproduced by general users.

A closely related method, called L2RW [1], adopts a similar meta-learning mechanism compared with ours. The major difference is that the weights are implicitly learned there, without an explicit weighting function. This, however, might lead to unstable weighting behavior during training and unavailability for generalization. In contrast, with the explicit yet simple Meta-Weight-Net, our method can learn the weight in a more stable way, as shown in Fig. 6, and can be easily generalized from a certain task to related other ones (see in the supplementary material).

**Other Methods for Class Imbalance.** Other methods for handling data imbalance include: [53, 54] tries to transfer the knowledge learned from major classes to minor classes. The metric learning based methods have also been developed to effectively exploit the tailed data to improve the generalization ability, e.g., triple-header loss [55] and range loss [56].

**Other Methods for Corrupted Labels.** For handling noisy label issue, multiple methods have been designed by correcting noisy labels to their true ones via a supplemental clean label inference step [11, 14, 57, 13, 21, 1, 15]. For example, GLC [15] proposed a loss correction approach to mitigate the effects of label noise on DNN classifiers. Other methods along this line include the Reed [58], Co-training [16], D2L [59] and S-Model [12].

## 4 Experimental Results

To evaluate the capability of the proposed algorithm, we implement experiments on data sets with class imbalance and noisy label issues, and real-world dataset with more complicated data bias.

### 4.1 Class Imbalance Experiments

We use Long-Tailed CIFAR dataset [60], that reduces the number of training samples per class according to an exponential function $n = n_i \mu^i$, where $i$ is the class index, $n_i$ is the original number of training images and $\mu \in (0, 1)$. The imbalance factor of a dataset is defined as the number of training samples in the largest class divided by the smallest. We trained ResNet-32 [61] with softmax cross-entropy loss by SGD with a momentum 0.9, a weight decay $5 \times 10^{-4}$, an initial learning rate 0.1. The learning rate of ResNet-32 is divided by 10 after 80 and 90 epoch (for a total 100 epochs), and the learning rate of WN-Net is fixed as $10^{-5}$. We randomly selected 10 images per class in validation set as the meta-data set. The compared methods include: 1) **BaseModel**, which uses a softmax cross-entropy loss to train ResNet-32 on the training set; 2) **Focal loss** [25] and **Class-Balanced** [60] represent the state-of-the-arts of the predefined sample reweighting techniques; 3) **Fine-tuning**, fine-tune the result of BaseModel on the meta-data set; 4) **L2RW** [1], which leverages an additional meta-dataset to adaptively assign weights on training samples.

Table 1 shows the classification accuracy of ResNet-32 on the test set and confusion matrices are displayed in Fig. 3 (more details are listed in the supplementary material). It can be observed that: 1) Our algorithm evidently outperforms other competing methods on datasets with class imbalance, showing its robustness in such data bias case; 2) When imbalance factor is 1, i.e., all classes are with same numbers of samples, fine-tuning runs best, and our method still attains a comparable performance; 3) When imbalance factor is 200 on long-tailed CIFAR-100, the smallest class has only two samples. An extra fine-tuning achieves performance gain, while our method still perform well in such extreme data bias.

To understand the weighing scheme of MW-Net, we depict the tendency curve of weight with respect to loss by the learned MW-Net in Fig. 1(d), which complies with the classical optimal weighting manner to such data bias. i.e., larger weights should be imposed on samples with relatively large losses, which are more likely to be minority class sample.

Table 1: Test accuracy (%) of ResNet-32 on long-tailed CIFAR-10 and CIFAR-100, and the best and the second best results are highlighted in **bold** and ***italic bold***, respectively.

| Dataset Name | Long-Tailed CIFAR-10 | | | | | | Long-Tailed CIFAR-100 | | | | | |
|---|---|---|---|---|---|---|---|---|---|---|---|---|
| Imbalance | 200 | 100 | 50 | 20 | 10 | 1 | 200 | 100 | 50 | 20 | 10 | 1 |
| BaseModel | 65.68 | 70.36 | 74.81 | 82.23 | 86.39 | 92.89 | 34.84 | 38.32 | 43.85 | 51.14 | 55.71 | 70.50 |
| Focal Loss | 65.29 | 70.38 | 76.71 | 82.76 | 86.66 | *93.03* | 35.62 | 38.41 | 44.32 | 51.95 | 55.78 | *70.52* |
| Class-Balanced | *68.89* | *74.57* | *79.27* | *84.36* | *87.49* | 92.89 | 36.23 | 39.60 | 45.32 | *52.59* | *57.99* | 70.50 |
| Fine-tuning | 66.08 | 71.33 | 77.42 | 83.37 | 86.42 | **93.23** | 38.22 | *41.83* | *46.40* | 52.11 | 57.44 | **70.72** |
| L2RW | 66.51 | 74.16 | 78.93 | 82.12 | 85.19 | 89.25 | 33.38 | 40.23 | 44.44 | 51.64 | 53.73 | 64.11 |
| Ours | **68.91** | **75.21** | **80.06** | **84.94** | **87.84** | 92.66 | *37.91* | **42.09** | **46.74** | **54.37** | **58.46** | 70.37 |

Table 2: Test accuracy comparison on CIFAR-10 and CIFAR-100 of WRN-28-10 with varying noise rates under uniform noise. Mean accuracy (±std) over 5 repetitions are reported ('—' means the method fails).

| Datasets / Noise Rate | | BaseModel | Reed-Hard | S-Model | Self-paced | Focal Loss | Co-teaching | D2L | Fine-tining | MentorNet | L2RW | GLC | Ours |
|---|---|---|---|---|---|---|---|---|---|---|---|---|---|
| CIFAR-10 | 0% | 95.60±0.22 | 94.38±0.14 | 83.79±0.11 | 90.81±0.34 | **95.70±0.15** | 88.67±0.25 | 94.64±0.33 | *95.65±0.15* | 94.35±0.42 | 92.38±0.10 | 94.30±0.19 | 94.52±0.25 |
| | 40% | 68.07±1.23 | 81.26±0.51 | 79.58±0.33 | 86.41±0.29 | 75.96±1.31 | 74.81±0.34 | 85.60±0.13 | 80.47±0.25 | 87.33±0.22 | 86.92±0.19 | *88.28±0.03* | **89.27±0.28** |
| | 60% | 53.12±3.03 | 73.53±1.54 | — | 53.10±1.78 | 51.87±1.19 | 73.06±0.25 | 68.02±0.41 | 78.75±2.40 | 82.80±1.35 | 82.24±0.36 | *83.49±0.24* | **84.07±0.33** |
| CIFAR-100 | 0% | 79.95±1.26 | 64.45±1.02 | 52.86±0.99 | 59.79±0.46 | **81.04±0.24** | 61.80±0.25 | 66.17±1.42 | *80.88±0.21* | 73.26±1.23 | 72.99±0.58 | 73.75±0.51 | 78.76±0.24 |
| | 40% | 51.11±0.42 | 51.27±1.18 | 42.12±0.99 | 46.31±2.45 | 51.19±0.46 | 46.20±0.15 | 52.10±0.97 | 52.49±0.74 | *61.39±3.99* | 60.79±0.91 | 61.31±0.22 | **67.73±0.26** |
| | 60% | 30.92±0.33 | 26.95±0.98 | — | 19.08±0.57 | 27.70±3.77 | 35.67±1.25 | 41.11±0.30 | 38.16±0.38 | 36.87±1.47 | 48.15±0.34 | *50.81±1.00* | **58.75±0.11** |

## 4.2 Corrupted Label Experiment

We study two settings of corrupted labels on the training set: 1) **Uniform noise.** The label of each sample is independently changed to a random class with probability $p$ following the same setting in [2]. 2) **Flip noise.** The label of each sample is independently flipped to similar classes with total probability $p$. In our experiments, we randomly select two classes as similar classes with equal probability. Two benchmark datasets are employed: CIFAR-10 and CIFAR-100 [62]. Both are popularly used for evaluation of noisy labels [59, 16].1000 images with clean labels in validation set are randomly selected as the meta-data set. We adopt a Wide ResNet-28-10 (WRN-28-10) [63] for uniform noise and ResNet-32 [61] for flip noise as our classifier network models[4].

The comparison methods include: **BaseModel**, referring to the similar classifier network utilized in our method, while directly trained on the biased training data; the robust learning methods **Reed** [58], **S-Model** [12] , **SPL** [26], **Focal Loss** [25], **Co-teaching** [16], **D2L** [59]; **Fine-tuning**, fine-tuning the result of **BaseModel** on the meta-data with clean labels to further enhance its performance; typical meta-learning methods **MentorNet** [21], **L2RW** [1], **GLC** [15]. We also trained the baseline network only on 1000 meta-images. The performance are evidently worse than the proposed method due to the neglecting of the knowledge underlying large amount of training samples. We thus have not involved its results in comparison.

All the baseline networks were trained using SGD with a momentum 0.9, a weight decay $5 \times 10^{-4}$ and an initial learning rate 0.1. The learning rate of classifier network is divided by 10 after 36 epoch and 38 epoch (for a total of 40 epoches) in uniform noise, and after 40 epoch and 50 epoch (for a total of 60 epoches) in flip noise. The learning rate of WN-Net is fixed as $10^{-3}$. We repeated the experiments 5 times with different random seeds for network initialization and label noise generation.

We report the accuracy averaged over 5 repetitions for each series of experiments and each competing method in Tables 2 and 3. It can be observed that our method gets the best performance across almost all datasets and all noise rates, except the second for 40% Flip noise. At 0% noise cases (unbiased ones), our method performs only slightly worse than the BaseModel. For other corrupted label cases, the superiority of our method is evident. Besides, it can be seen that the performance gaps between ours and all other competing methods increase as the noise rate is increased from 40% to 60% under uniform noise. Even with 60% label noise, our method can still obtain a relatively high classification accuracy, and attains more than 15% accuracy gain compared with the second best result for CIFAR100 dataset, which indicates the robustness of our methods in such cases.

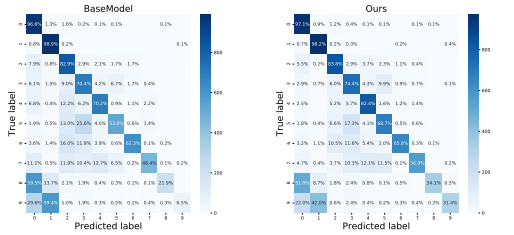
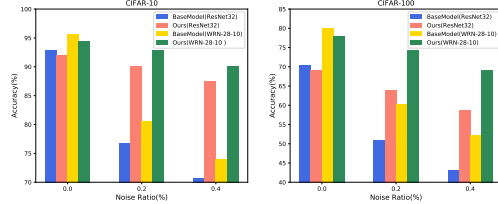

Figure 3: Confusion matrices for the Basemodel and ours on long-tailed CIFAR-10 with imbalance factors 200.

Figure 4: Performance comparison for different classifier networks (WRN-28-10 and ResNet32) under CIFAR flip noise.

Table 3: Test accuracy comparison on CIFAR-10 and CIFAR-100 of ResNet-32 with varying noise rates under flip noise.

| Datasets / Noise Rate | | BaseModel | Reed-Hard | S-Model | Self-paced | Focal Loss | Co-teaching | D2L | Fine-tining | MentorNet | L2RW | GLC | Ours |
|---|---|---|---|---|---|---|---|---|---|---|---|---|---|
| CIFAR-10 | 0% | 92.89±0.32 | 92.31±0.25 | 83.61±0.13 | 88.52±0.21 | 93.03±0.16 | 89.87±0.10 | 92.02±0.14 | 93.23±0.23 | 92.13±0.30 | 89.25±0.37 | 91.02±0.20 | 92.04±0.15 |
| | 20% | 76.83±2.30 | 88.28±0.36 | 79.25±0.30 | 87.03±0.34 | 86.45±0.19 | 82.83±0.85 | 87.66±0.40 | 82.47±3.64 | 86.36±0.31 | 87.86±0.36 | 89.68±0.33 | 90.33±0.61 |
| | 40% | 70.77±2.31 | 81.06±0.76 | 75.73±0.32 | 81.63±0.52 | 80.45±0.97 | 75.41±0.21 | 83.89±0.46 | 74.07±1.56 | 81.76±0.28 | 85.66±0.51 | 88.92±0.24 | 87.54±0.23 |
| CIFAR-100 | 0% | 70.50±0.12 | 69.02±0.32 | 51.46±0.20 | 67.55±0.27 | 70.02±0.53 | 63.31±0.05 | 68.11±0.26 | 70.72±0.22 | 70.24±0.21 | 64.11±1.09 | 65.42±0.23 | 70.11±0.33 |
| | 20% | 50.86±0.27 | 60.27±0.76 | 45.45±0.25 | 63.63±0.30 | 61.87±0.30 | 54.13±0.55 | 63.48±0.53 | 56.98±0.50 | 61.97±0.47 | 57.47±1.16 | 63.07±0.53 | 64.22±0.28 |
| | 40% | 43.01±1.16 | 50.40±1.01 | 43.81±0.15 | 53.51±0.53 | 54.13±0.40 | 44.85±0.81 | 51.83±0.33 | 46.37±0.25 | 52.66±0.56 | 50.98±1.55 | 62.22±0.62 | 58.64±0.47 |

Fig. 4 shows the performance comparison between WRN-28-10 and ResNet32 under fixed flip noise setting. We can observe that the performance gains for our method and BaseModel between two networks takes the almost same value. It implies that the performance improvement of our method is not dependent on the selection of the classifier network architectures.

As shown in Fig. 1(e), the shape of the learned weight function depicts as monotonic decreasing, complying with the traditional optimal setting to this bias condition, i.e., imposing smaller weights on samples with relatively large losses to suppress the effect of corrupted labels. Furthermore, we plot the weight distribution of clean and noisy training samples in Fig. 5. It can be seen that almost all large weights belongs to clean samples, and the noisy samples's weights are smaller than that of clean samples, which implies that the trained Meta-Weight-Net can distinguish clean and noisy images.

Fig. 6 plots the weight variation along with training epochs under 40% noise on CIFAR10 dataset of our method and L2RW. $y$-axis denotes the differences of weights calculated between adjacent epochs, and $x$-axis denotes the number of epochs. Ten noisy samples are randomly chosen to compute their mean curve, surrounded by the region illustrating the standard deviations calculated on these samples in the corresponding epoch. It is seen that the weight by our method is continuously changed, gradually stable along iterations, and finally converges. As a comparison, the weight during the learning process of L2RW fluctuates relatively more wildly. This could explain the consistently better performance of our method as compared with this competing method.

## 4.3 Experiments on Clothing1M

To verify the effectiveness of the proposed method on real-world data, we conduct experiments on the Clothing1M dataset [64], containing 1 million images of clothing obtained from online shopping websites that are with 14 categories, e.g., T-shirt, Shirt, Knitwear. The labels are generated by using surrounding texts of the images provided by the sellers, and therefore contain many errors. We use the 7k clean data as the meta dataset. Following the previous works [65, 66], we used ResNet-50 pre-trained on ImageNet. For preprocessing, we resize the image to $256 \times 256$, crop the middle $224 \times 224$ as input, and perform normalization. We used SGD with a momentum 0.9, a weight decay $10^{-3}$, and an initial learning rate 0.01, and batch size 32. The learning rate of ResNet-50 is divided by 10 after 5 epoch (for a total 10 epoch), and the learning rate of WN-Net is fixed as $10^{-3}$.

The results are summarized in Table. 4. which shows that the proposed method achieves the best performance. Fig. 1(f) plots the tendency curve of the learned MW-Net function, which reveals abundant data insights. Specifically, when the loss is with relatively small values, the weighting function inclines to increase with loss, meaning that it tends to more emphasize hard margin samples with informative knowledge for classification; while when the loss gradually changes large, the

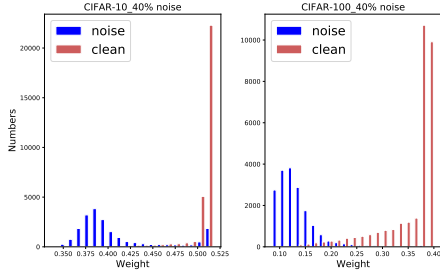

Figure 5: Sample weight distribution on training data under 40% uniform noise experiments.

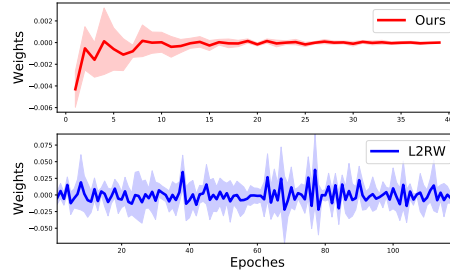

Figure 6: Weight variation curves under 40% uniform noise experiment on CIFAR10 dataset.

Table 4: Classification accuracy (%) of all competing methods on the Clothing1M test set.

| # | Method | Accuracy | # | Method | Accuracy |
|---|--------|----------|---|--------|----------|
| 1 | Cross Entropy | 68.94 | 5 | Joint Optimization [66] | 72.23 |
| 2 | Bootstrapping [58] | 69.12 | 6 | LCCN [67] | 73.07 |
| 3 | Forward [65] | 69.84 | 7 | MLNT [68] | 73.47 |
| 4 | S-adaptation [12] | 70.36 | 8 | Ours | 73.72 |

weighting function begins to monotonically decrease, implying that it tends to suppress noise labels samples with relatively large loss values. Such complicated essence cannot be finely delivered by conventional weight functions.

## 5 Conclusion

We have proposed a novel meta-learning method for adaptively extracting sample weights to guarantee robust deep learning in the presence of training data bias. Compared with current reweighting methods that require to manually set the form of weight functions, the new method is able to yield a rational one directly from data. The working principle of our algorithm can be well explained and the procedure of our method can be easily reproduced ( Appendix A provide the Pytorch implement of our algorithm (less than 30 lines of codes)), and the completed training code is avriable at `https://github.com/xjtushujun/meta-weight-net`.). Our empirical results show that the propose method can perform superior in general data bias cases, like class imbalance, corrupted labels, and more complicated real cases. Besides, such an adaptive weight learning approach is hopeful to be employed to other weight setting problems in machine learning, like ensemble methods and multi-view learning.

### Acknowledgments

This research was supported by the China NSFC projects under contracts 61661166011, 11690011, 61603292, 61721002,U1811461. The authors would also like to thank anonymous reviewers for their constructive suggestions on improving the paper, especially on the proofs and theoretical analysis of our paper.

## Footnotes

[1]We call the training data biased when they are generated from a joint sample-label distribution deviating from the distribution of evaluation/test set[1].

[2]Notice that $\Theta$ here is a variable instead of a quantity, which makes $\hat{\mathbf{w}}^t(\Theta)$ a function of $\Theta$ and the gradient in Eq. (4) be able to be computed.

[3]Derivation can be found in supplementary materials.

[4]We have tried different classifier network architectures as classifier networks under each noise setting to show our algorithm is suitable to different deep learning architectures. We show this effect in Fig.4, verifying the consistently good performance of our method in two classifier network settings.

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
