[Supplementary Material]

# Meta-Weight-Net: Learning an Explicit Mapping For Sample Weighting

**Jun Shu**
Xi'an Jiaotong University
xjtushujun@gmail.com

**Qi Xie**
Xi'an Jiaotong University
xq.liwu@stu.xjtu.edu.cn

**Lixuan Yi**
Xi'an Jiaotong University
yilixuan@stu.xjtu.edu.cn

**Qian Zhao**
Xi'an Jiaotong University
timmy.zhaoqian@mail.xjtu.edu.cn

**Sanping Zhou**
Xi'an Jiaotong University
sanpingzhou@stu.xjtu.edu.cn

**Zongben Xu**
Xi'an Jiaotong University
zbxu@mail.xjtu.edu.cn

**Deyu Meng**[*]
Xi'an Jiaotong University
dymeng@mail.xjtu.edu.cn

## Abstract

Current deep neural networks (DNNs) can easily overfit to biased training data with corrupted labels or class imbalance. Sample re-weighting strategy is commonly used to alleviate this issue by designing a weighting function mapping from training loss to sample weight, and then iterating between weight recalculating and classifier updating. Current approaches, however, need manually pre-specify the weighting function as well as its additional hyper-parameters. It makes them fairly hard to be generally applied in practice due to the significant variation of proper weighting schemes relying on the investigated problem and training data. To address this issue, we propose a method capable of adaptively learning an explicit weighting function directly from data. The weighting function is an MLP with one hidden layer, constituting a universal approximator to almost any continuous functions, making the method able to fit a wide range of weighting functions including those assumed in conventional research. Guided by a small amount of unbiased meta-data, the parameters of the weighting function can be finely updated simultaneously with the learning process of the classifiers. Synthetic and real experiments substantiate the capability of our method for achieving proper weighting functions in class imbalance and noisy label cases, fully complying with the common settings in traditional methods, and more complicated scenarios beyond conventional cases. This naturally leads to its better accuracy than other state-of-the-art methods. Source code is available at `https://github.com/xjtushujun/meta-weight-net`.

## 1 Introduction

DNNs have recently obtained impressive good performance on various applications due to their powerful capacity for modeling complex input patterns. However, DNNs can easily overfit to biased training data[1], like those containing corrupted labels [2] or with class imbalance[3], leading to

---

[*]Corresponding author.

[1]We call the training data biased when they are generated from a joint sample-label distribution deviating from the distribution of evaluation/test set[1].

(a) Weight function in focal loss    (b) Weight function in SPL    (c) Meta-Weight-Net architecture

(d) MW-Net function learned in class imbalance case    (e) MW-Net function learned in corrupter labels case    (f) MW-Net function learned in real Clothing1M dataset

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

[1]Since our method is inspired by L2RW [1], we compare this method, as well as the BaseModel, for better illustration. The uniform noise experiment setting is the same as L2RW [1], and thus the iteration steps of BaseModel and L2RW follows the original setting.

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

# Appendix

## A  Pytorch codes of our algorithm

The following is the Pytorch codes of our algorithm (core code is less than 30 lines), and the completed training code is avriable at `https://github.com/xjtushujun/meta-weight-net`.

```python
def norm_func(v_lambda):
  norm_c = torch.sum(v_lambda)
  if norm_c != 0:
    v_lambda_norm = v_lambda / norm_c
  else:
    v_lambda_norm = v_lambda
  return  v_lambda_norm

optimizer_a = torch.optim.SGD(model.params(), args.lr, momentum=args.
                                  momentum, nesterov=args.nesterov,
                                  weight_decay=args.weight_decay)
optimizer_c = torch.optim.SGD(vnet.params(), 1e-3, momentum=args.
                                  momentum, nesterov=args.nesterov,
                                  weight_decay=args.weight_decay)

for iters in range(num_iters):
  adjust_learning_rate(optimizer_a, iters + 1)
  model.train()
  data, target = next(iter(train_loader))
  data, target = data.to(device), target.to(device)
  meta_model.load_state_dict(model.state_dict())
  y_f_hat = meta_model(data)
  cost = F.cross_entropy(y_f_hat, target_var, reduce=False)
  cost_v = torch.reshape(cost, (len(cost), 1))
  v_lambda = vnet(cost_v.data)
  v_lambda_norm = norm_func(v_lambda)
  l_f_meta = torch.sum(cost_v * v_lambda_norm)
  meta_model.zero_grad()
  grads = torch.autograd.grad(l_f_meta,(meta_model.params()),
                                  create_graph=True)
  meta_model.update_params(lr_inner=meta_lr,source_params=grads)

  data_meta,target_meta = next(iter(train_meta_loader))
  data_meta,target_meta = data_meta.to(device),target_meta.to(device)
  y_g_hat = meta_model(data_meta)
  l_g_meta = F.cross_entropy(y_g_hat, target_meta)
  optimizer_c.zero_grad()
  l_g_meta.backward()
  optimizer_c.step()

  y_f = model(data)
  cost_w = F.cross_entropy(y_f, target, reduce=False)
  cost_v = torch.reshape(cost_w, (len(cost_w), 1))
  with torch.no_grad():
    w_new = vnet(cost_v)
  w_v = norm_func(w_new)
  l_f = torch.sum(cost_v * w_v)
  optimizer_a.zero_grad()
  l_f.backward()
  optimizer_a.step()
```

## B  Derivation of the Weighting Scheme in Meta-Weight-Net

Recall the update equation of the parameters of Meta-Weight-Net as follows:

$$\Theta^{(t+1)} = \Theta^{(t)} - \beta \frac{1}{m} \sum_{i=1}^{m} \nabla_{\Theta} L_i^{meta}(\hat{\mathbf{w}}^{(t)}(\Theta)) \Big|_{\Theta^{(t)}}. \tag{9}$$

The computation of Eq. (9) in the paper by backpropagation can be understood by the following derivation:

$$
\begin{aligned}
&\frac{1}{m} \sum_{i=1}^{m} \nabla_{\Theta} L_i^{meta}(\hat{\mathbf{w}}^{(t)}(\Theta)) \Big|_{\Theta^{(t)}} \\
&= \frac{1}{m} \sum_{i=1}^{m} \frac{\partial L_i^{meta}(\hat{\mathbf{w}})}{\partial \hat{\mathbf{w}}(\Theta)} \Big|_{\hat{\mathbf{w}}^{(t)}} \sum_{j=1}^{n} \frac{\partial \hat{\mathbf{w}}^{(t)}(\Theta)}{\partial \mathcal{V}(L_j^{train}(\mathbf{w}^{(t)}); \Theta)} \frac{\partial \mathcal{V}(L_j^{train}(\mathbf{w}^{(t)}); \Theta)}{\partial \Theta} \Big|_{\Theta^{(t)}} \\
&= \frac{-\alpha}{n*m} \sum_{i=1}^{m} \frac{\partial L_i^{meta}(\hat{\mathbf{w}})}{\partial \hat{\mathbf{w}}} \Big|_{\hat{\mathbf{w}}^{(t)}} \sum_{j=1}^{n} \frac{\partial L_j^{train}(\mathbf{w})}{\partial \mathbf{w}} \Big|_{\mathbf{w}^{(t)}} \frac{\partial \mathcal{V}(L_j^{train}(\mathbf{w}^{(t)}); \Theta)}{\partial \Theta} \Big|_{\Theta^{(t)}} \\
&= \frac{-\alpha}{n} \sum_{j=1}^{n} \left( \frac{1}{m} \sum_{i=1}^{m} \frac{\partial L_i^{meta}(\hat{\mathbf{w}})}{\partial \hat{\mathbf{w}}} \Big|_{\hat{\mathbf{w}}^{(t)}}^{T} \frac{\partial L_j^{train}(\mathbf{w})}{\partial \mathbf{w}} \Big|_{\mathbf{w}^{(t)}} \right) \frac{\partial \mathcal{V}(L_j^{train}(\mathbf{w}^{(t)}); \Theta)}{\partial \Theta} \Big|_{\Theta^{(t)}}. \tag{10}
\end{aligned}
$$

Let

$$G_{ij} = \frac{\partial L_i^{meta}(\hat{\mathbf{w}})}{\partial \hat{\mathbf{w}}} \Big|_{\hat{\mathbf{w}}^{(t)}}^{T} \frac{\partial L_j^{train}(\mathbf{w})}{\partial \mathbf{w}} \Big|_{\mathbf{w}^{(t)}}, \tag{11}$$

by substituting Eq. (10) into Eq. (9), we can get:

$$\Theta^{(t+1)} = \Theta^{(t)} + \frac{\alpha\beta}{n} \sum_{j=1}^{n} \left( \frac{1}{m} \sum_{i=1}^{m} G_{ij} \right) \frac{\partial \mathcal{V}(L_j^{train}(\mathbf{w}^{(t)}); \Theta)}{\partial \Theta} \Big|_{\Theta^{(t)}}. \tag{12}$$

## C  Convergence Proof of Our Method

This section provides the proofs for Theorems 1 and 2 in the paper.

Suppose that we have a small amount of meta (validation) dataset with $M$ samples $\{(x_i^{(m)}, y_i^{(m)}), 1 \leq i \leq M\}$ with clean labels, and the overall meta loss is,

$$\mathcal{L}^{meta}(\mathbf{w}^*(\Theta)) = \frac{1}{M} \sum_{i=1}^{M} L_i^{meta}(\mathbf{w}^*(\Theta)), \tag{13}$$

where $\mathbf{w}^*$ is the parameter of the classifier network, and $\Theta$ is the parameter of the Meta-Weight-Net. Let's suppose we have another $N$ training data, $\{(x_i, y_i), 1 \leq i \leq N\}$, where $M \ll N$, and the overall training loss is,

$$\mathcal{L}^{train}(\mathbf{w}; \Theta) = \frac{1}{N} \sum_{i=1}^{N} \mathcal{V}(L_i^{train}(\mathbf{w}); \Theta) L_i^{train}(\mathbf{w}). \tag{14}$$

**Lemma 1.** *Suppose the meta loss function is Lipschitz smooth with constant $L$, and $\mathcal{V}(\cdot)$ is differential with a $\delta$-bounded gradient and twice differential with its Hessian bounded by $\mathcal{B}$, and the loss function $\ell$ have $\rho$-bounded gradients with respect to training/meta data. Then the gradient of $\Theta$ with respect to meta loss is Lipschitz continuous.*

*Proof.* The gradient of $\Theta$ with respect to meta loss can be written as:

$$
\begin{aligned}
&\nabla_{\Theta} L_i^{meta}(\hat{\mathbf{w}}^{(t)}(\Theta)) \Big|_{\Theta^{(t)}} \\
&= \frac{-\alpha}{n} \sum_{j=1}^{n} \left( \frac{\partial L_i^{meta}(\hat{\mathbf{w}})}{\partial \hat{\mathbf{w}}} \Big|_{\hat{\mathbf{w}}^{(t)}}^{T} \frac{\partial L_j^{train}(\mathbf{w})}{\partial \mathbf{w}} \Big|_{\mathbf{w}^{(t)}} \right) \frac{\partial \mathcal{V}(L_j^{train}(\mathbf{w}^{(t)}); \Theta)}{\partial \Theta} \Big|_{\Theta^{(t)}}. \tag{15}
\end{aligned}
$$

Let $\mathcal{V}_j(\Theta) = \mathcal{V}(L_j^{train}(\mathbf{w}^{(t)}); \Theta)$ and $G_{ij}$ being defined in Eq.(11). Taking gradient of $\Theta$ in both sides of Eq.(15), we have

$$\nabla_{\Theta^2}^2 L_i^{meta}(\hat{\mathbf{w}}^{(t)}(\Theta))\Big|_{\Theta^{(t)}} = \frac{-\alpha}{n} \sum_{j=1}^{n} \left[ \frac{\partial}{\partial \Theta} (G_{ij}) \Big|_{\Theta^{(t)}} \frac{\partial \mathcal{V}_j(\Theta)}{\partial \Theta} \Big|_{\Theta^{(t)}} + (G_{ij}) \frac{\partial^2 \mathcal{V}_j(\Theta)}{\partial \Theta^2} \Big|_{\Theta^{(t)}} \right].$$

For the first term in the right hand side, we have that

$$\left\| \frac{\partial}{\partial \Theta}(G_{ij})\Big|_{\Theta^{(t)}} \frac{\partial \mathcal{V}_j(\Theta)}{\partial \Theta}\Big|_{\Theta^{(t)}} \right\| \leq \delta \left\| \frac{\partial}{\partial \hat{\mathbf{w}}} \left( \frac{\partial L_i^{meta}(\hat{\mathbf{w}})}{\partial \Theta}\Big|_{\Theta^{(t)}} \right)^T \Big|_{\hat{\mathbf{w}}^{(t)}} \frac{\partial L_j^{train}(\mathbf{w})}{\partial \mathbf{w}}\Big|_{\mathbf{w}^{(t)}} \right\|$$

$$= \delta \left\| \frac{\partial}{\partial \hat{\mathbf{w}}} \left( \frac{\partial L_i^{meta}(\hat{\mathbf{w}})}{\partial \hat{\mathbf{w}}}\Big|_{\hat{\mathbf{w}}^{(t)}} \frac{-\alpha}{n} \sum_{k=1}^{n} \frac{\partial L_k^{train}(\mathbf{w})}{\partial \mathbf{w}}\Big|_{\mathbf{w}^{(t)}} \frac{\partial \mathcal{V}_k(\Theta)}{\partial \Theta}\Big|_{\Theta^{(t)}} \right)^T \Big|_{\hat{\mathbf{w}}^{(t)}} \frac{\partial L_j^{train}(\mathbf{w})}{\partial \mathbf{w}}\Big|_{\mathbf{w}^{(t)}} \right\|$$

$$= \delta \left\| \left( \frac{\partial^2 L_i^{meta}(\hat{\mathbf{w}})}{\partial \hat{\mathbf{w}}^2}\Big|_{\hat{\mathbf{w}}^{(t)}} \frac{-\alpha}{n} \sum_{k=1}^{n} \frac{\partial L_k^{train}(\mathbf{w})}{\partial \mathbf{w}}\Big|_{\mathbf{w}^{(t)}} \frac{\partial \mathcal{V}_k(\Theta)}{\partial \Theta}\Big|_{\Theta^{(t)}} \right)^T \Big|_{\hat{\mathbf{w}}^{(t)}} \frac{\partial L_j^{train}(\mathbf{w})}{\partial \mathbf{w}} \right\| \leq \alpha L \rho^2 \delta^2, \quad (16)$$

since $\left\| \frac{\partial^2 L_i^{meta}(\hat{\mathbf{w}})}{\partial \hat{\mathbf{w}}^2}\Big|_{\hat{\mathbf{w}}^{(t)}} \right\| \leq L$, $\left\| \frac{\partial L_j^{train}(\mathbf{w})}{\partial \mathbf{w}}\Big|_{\mathbf{w}^{(t)}} \right\| \leq \rho$, $\left\| \frac{\partial \mathcal{V}_j(\Theta)}{\partial \Theta}\Big|_{\Theta^{(t)}} \right\| \leq \delta$. And for the second term we have

$$\left\| (G_{ij}) \frac{\partial^2 \mathcal{V}_j(\Theta)}{\partial \Theta^2}\Big|_{\Theta^{(t)}} \right\| = \left\| \frac{\partial L_i^{meta}(\hat{\mathbf{w}})}{\partial \hat{\mathbf{w}}}\Big|_{\hat{\mathbf{w}}^{(t)}}^T \frac{\partial L_j^{train}(\mathbf{w})}{\partial \mathbf{w}}\Big|_{\mathbf{w}^{(t)}} \frac{\partial^2 \mathcal{V}_j(\Theta)}{\partial \Theta^2}\Big|_{\Theta^{(t)}} \right\| \leq \mathcal{B} \rho^2, \quad (17)$$

since $\left\| \frac{\partial L_i^{meta}(\hat{\mathbf{w}})}{\partial \hat{\mathbf{w}}}\Big|_{\hat{\mathbf{w}}^{(t)}}^T \right\| \leq \rho$, $\left\| \frac{\partial^2 \mathcal{V}_j(\Theta)}{\partial \Theta^2}\Big|_{\Theta^{(t)}} \right\| \leq \mathcal{B}$. Combining the above two inequalities Eq.(16) (17), we have

$$\left\| \nabla_{\Theta^2}^2 L_i^{meta}(\hat{\mathbf{w}}^{(t)}(\Theta))\Big|_{\Theta^{(t)}} \right\| \leq \alpha \rho^2 (\alpha L \delta^2 + \mathcal{B}). \quad (18)$$

Define $L_V = \alpha \rho^2 (\alpha L \delta^2 + \mathcal{B})$, based on Lagrange mean value theorem, we have:

$$\| \nabla \mathcal{L}^{meta}(\hat{\mathbf{w}}^{(t)}(\Theta_1)) - \nabla \mathcal{L}^{meta}(\hat{\mathbf{w}}^{(t)}(\Theta_2)) \| \leq L_V \| \Theta_1 - \Theta_2 \|, \quad for \; all \; \Theta_1, \Theta_2, \quad (19)$$

where $\nabla \mathcal{L}^{meta}(\hat{\mathbf{w}}^{(t)}(\Theta_1)) = \nabla_\Theta L_i^{meta}(\hat{\mathbf{w}}^{(t)}(\Theta))\Big|_{\Theta_1}$. $\qquad \square$

**Theorem 1.** *Suppose the loss function $\ell$ is Lipschitz smooth with constant $L$, and $\mathcal{V}(\cdot)$ is differential with a $\delta$-bounded gradient and twice differential with its Hessian bounded by $\mathcal{B}$, and the loss function $\ell$ have $\rho$-bounded gradients with respect to training/meta data. Let the learning rate $\alpha_t$ satisfies $\alpha_t = \min\{1, \frac{k}{T}\}$, for some $k > 0$, such that $\frac{k}{T} < 1$, and $\beta_t, 1 \leq t \leq N$ is a monotone descent sequence, $\beta_1 = \min\{\frac{1}{L}, \frac{c}{\sigma\sqrt{T}}\}$ for some $c > 0$, such that $\frac{\sigma\sqrt{T}}{c} \geq L$ and $\sum_{t=1}^{\infty} \beta_t \leq \infty, \sum_{t=1}^{\infty} \beta_t^2 \leq \infty$. Then Meta-Weight-Net can achieve $\mathbb{E}[\| \nabla \mathcal{L}^{meta}(\hat{\mathbf{w}}^{(t)}(\Theta^{(t)})) \|_2^2] \leq \epsilon$ in $\mathcal{O}(1/\epsilon^2)$ steps. More specifically,*

$$\min_{0 \leq t \leq T} \mathbb{E}[\| \nabla \mathcal{L}^{meta}(\hat{\mathbf{w}}^{(t)}(\Theta^{(t)})) \|_2^2] \leq \mathcal{O}(\frac{C}{\sqrt{T}}), \quad (20)$$

*where $C$ is some constant independent of the convergence process, $\sigma$ is the variance of drawing uniformly mini-batch sample at random.*

*Proof.* The update of $\Theta$ in each iteration is as follows:

$$\Theta^{(t+1)} = \Theta^{(t)} - \beta \frac{1}{m} \sum_{i=1}^{m} \nabla_\Theta L_i^{meta}(\hat{\mathbf{w}}^{(t)}(\Theta))\Big|_{\Theta^{(t)}}. \quad (21)$$

This can be written as:

$$\Theta^{(t+1)} = \Theta^{(t)} - \beta_t \nabla \mathcal{L}^{meta}(\hat{\mathbf{w}}^{(t)}(\Theta^{(t)}))\Big|_{\Xi_t}. \quad (22)$$

Since the mini-batch $\Xi_t$ is drawn uniformly from the entire data set, we can rewrite the update equationp as:

$$\Theta^{(t+1)} = \Theta^{(t)} - \beta_t [\nabla \mathcal{L}^{meta}(\hat{\mathbf{w}}^{(t)}(\Theta^{(t)})) + \xi^{(t)}], \quad (23)$$

where $\xi^{(t)} = \nabla\mathcal{L}^{meta}(\hat{\mathbf{w}}^{(t)}(\Theta^{(t)}))\big|_{\Xi_t} - \nabla\mathcal{L}^{meta}(\hat{\mathbf{w}}^{(t)}(\Theta^{(t)}))$. Note that $\xi^{(t)}$ are i.i.d random variable with finite variance, since $\Xi_t$ are drawn i.i.d with a finite number of samples. Furthermore, $\mathbb{E}[\xi^{(t)}] = 0$, since samples are drawn uniformly at random. Observe that

$$\mathcal{L}^{meta}(\hat{\mathbf{w}}^{(t+1)}(\Theta^{(t+1)})) - \mathcal{L}^{meta}(\hat{\mathbf{w}}^{(t)}(\Theta^{(t)}))$$
$$= \left\{\mathcal{L}^{meta}(\hat{\mathbf{w}}^{(t+1)}(\Theta^{(t+1)})) - \mathcal{L}^{meta}(\hat{\mathbf{w}}^{(t)}(\Theta^{(t+1)}))\right\} + \left\{\mathcal{L}^{meta}(\hat{\mathbf{w}}^{(t)}(\Theta^{(t+1)})) - \mathcal{L}^{meta}(\hat{\mathbf{w}}^{(t)}(\Theta^{(t)}))\right\}. \quad (24)$$

By Lipschitz smoothness of meta loss function, we have

$$\mathcal{L}^{meta}(\hat{\mathbf{w}}^{(t+1)}(\Theta^{(t+1)})) - \mathcal{L}^{meta}(\hat{\mathbf{w}}^{(t)}(\Theta^{(t+1)}))$$
$$\leq \langle\nabla\mathcal{L}^{meta}(\hat{\mathbf{w}}^{(t)}(\Theta^{(t+1)})), \hat{\mathbf{w}}^{(t+1)}(\Theta^{(t+1)}) - \hat{\mathbf{w}}^{(t)}(\Theta^{(t+1)})\rangle + \frac{L}{2}\|\hat{\mathbf{w}}^{(t+1)}(\Theta^{(t+1)}) - \hat{\mathbf{w}}^{(t)}(\Theta^{(t+1)})\|_2^2$$

Since $\hat{\mathbf{w}}^{(t+1)}(\Theta^{(t+1)}) - \hat{\mathbf{w}}^{(t)}(\Theta^{(t+1)}) = -\alpha_t\frac{1}{n}\sum_{i=1}^{n}\mathcal{V}(L_i^{train}(\mathbf{w}^{(t+1)});\Theta^{(t+1)})\nabla_{\mathbf{w}}L_i^{train}(\mathbf{w})\big|_{\mathbf{w}^{(t+1)}}$ according to Eq.(3),(5), we have

$$\|\mathcal{L}^{meta}(\hat{\mathbf{w}}^{(t+1)}(\Theta^{(t+1)})) - \mathcal{L}^{meta}(\hat{\mathbf{w}}^{(t)}(\Theta^{(t+1)}))\| \leq \alpha_t\rho^2 + \frac{L\alpha_t^2}{2}\rho^2 = \alpha_t\rho^2(1 + \frac{\alpha_t L}{2}) \quad (25)$$

Since $\left\|\frac{\partial L_j^{train}(\mathbf{w})}{\partial\mathbf{w}}\big|_{\mathbf{w}^{(t)}}\right\| \leq \rho, \left\|\frac{\partial L_i^{meta}(\hat{\mathbf{w}})}{\partial\hat{\mathbf{w}}}\big|_{\hat{\mathbf{w}}^{(t)}}^T\right\| \leq \rho.$

By Lipschitz continuity of $\nabla\mathcal{L}^{meta}(\hat{\mathbf{w}}^{(t)}(\Theta))$ according to Lemma 1, we can obtain the following:

$$\mathcal{L}^{meta}(\hat{\mathbf{w}}^{(t)}(\Theta^{(t+1)})) - \mathcal{L}^{meta}(\hat{\mathbf{w}}^{(t)}(\Theta^{(t)})) \leq \langle\nabla\mathcal{L}^{meta}(\hat{\mathbf{w}}^{(t)}(\Theta^{(t)})), \Theta^{(t+1)} - \Theta^{(t)}\rangle + \frac{L}{2}\|\Theta^{(t+1)} - \Theta^{(t)}\|_2^2$$
$$= \langle\nabla\mathcal{L}^{meta}(\hat{\mathbf{w}}^{(t)}(\Theta^{(t)})), -\beta_t[\nabla\mathcal{L}^{meta}(\hat{\mathbf{w}}^{(t)}(\Theta^{(t)})) + \xi^{(t)}]\rangle + \frac{L\beta_t^2}{2}\|\nabla\mathcal{L}^{meta}(\hat{\mathbf{w}}^{(t)}(\Theta^{(t)})) + \xi^{(t)}\|_2^2$$
$$= -(\beta_t - \frac{L\beta_t^2}{2})\|\nabla\mathcal{L}^{meta}(\hat{\mathbf{w}}^{(t)}(\Theta^{(t)}))\|_2^2 + \frac{L\beta_t^2}{2}\|\xi^{(t)}\|_2^2 - (\beta_t - L\beta_t^2)\langle\nabla\mathcal{L}^{meta}(\hat{\mathbf{w}}^{(t)}(\Theta^{(t)})), \xi^{(t)}\rangle.$$

Thus Eq.(35) satifies

$$\mathcal{L}^{meta}(\hat{\mathbf{w}}^{(t+1)}(\Theta^{(t+1)})) - \mathcal{L}^{meta}(\hat{\mathbf{w}}^{(t)}(\Theta^{(t)})) \leq \alpha_t\rho^2(1 + \frac{\alpha_t L}{2}) - (\beta_t - \frac{L\beta_t^2}{2})$$
$$\|\nabla\mathcal{L}^{meta}(\hat{\mathbf{w}}^{(t)}(\Theta^{(t)}))\|_2^2 + \frac{L\beta_t^2}{2}\|\xi^{(t)}\|_2^2 - (\beta_t - L\beta_t^2)\langle\nabla\mathcal{L}^{meta}(\hat{\mathbf{w}}^{(t)}(\Theta^{(t)})), \xi^{(t)}\rangle. \quad (26)$$

Rearranging the terms, we can obtain

$$(\beta_t - \frac{L\beta_t^2}{2})\|\nabla\mathcal{L}^{meta}(\hat{\mathbf{w}}^{(t)}(\Theta^{(t)}))\|_2^2 \leq \alpha_t\rho^2(1 + \frac{\alpha_t L}{2}) + \mathcal{L}^{meta}(\hat{\mathbf{w}}^{(t)}(\Theta^{(t)})) - \mathcal{L}^{meta}(\hat{\mathbf{w}}^{(t+1)}(\Theta^{(t+1)}))$$
$$+ \frac{L\beta_t^2}{2}\|\xi^{(t)}\|_2^2 - (\beta_t - L\beta_t^2)\langle\nabla\mathcal{L}^{meta}(\hat{\mathbf{w}}^{(t)}(\Theta^{(t)})), \xi^{(t)}\rangle. \quad (27)$$

Summing up the above inequalities and rearranging the terms, we can obtain

$$\sum_{t=1}^{T}(\beta_t - \frac{L\beta_t^2}{2})\|\nabla\mathcal{L}^{meta}(\hat{\mathbf{w}}^{(t)}(\Theta^{(t)}))\|_2^2 \leq \mathcal{L}^{meta}(\hat{\mathbf{w}}^{(1)})(\Theta^{(1)}) - \mathcal{L}^{meta}(\hat{\mathbf{w}}^{(T+1)}(\Theta^{(T+1)}))$$
$$+ \sum_{t=1}^{T}\alpha_t\rho^2(1 + \frac{\alpha_t L}{2}) - \sum_{t=1}^{T}(\beta_t - L\beta_t^2)\langle\nabla\mathcal{L}^{meta}(\hat{\mathbf{w}}^{(t)}(\Theta^{(t)})), \xi^{(t)}\rangle + \frac{L}{2}\sum_{t=1}^{T}\beta_t^2\|\xi^{(t)}\|_2^2$$
$$\leq \mathcal{L}^{meta}(\hat{\mathbf{w}}^{(1)}(\Theta^{(1)})) + \sum_{t=1}^{T}\alpha_t\rho^2(1 + \frac{\alpha_t L}{2}) - \sum_{t=1}^{T}(\beta_t - L\beta_t^2)\langle\nabla\mathcal{L}^{meta}(\hat{\mathbf{w}}^{(t)}(\Theta^{(t)})), \xi^{(t)}\rangle + \frac{L}{2}\sum_{t=1}^{T}\beta_t^2\|\xi^{(t)}\|_2^2, \quad (28)$$

Taking expectations with respect to $\xi^{(N)}$ on both sides of Eq. 28, we can then obtain:

$$\sum_{t=1}^{T}(\beta_t - \frac{L\beta_t^2}{2})\mathbb{E}_{\xi^{(N)}}\|\nabla\mathcal{L}^{meta}(\hat{\mathbf{w}}^{(t)}(\Theta^{(t)}))\|_2^2 \leq \mathcal{L}^{meta}(\hat{\mathbf{w}}^{(1)}(\Theta^{(1)})) + \sum_{t=1}^{T}\alpha_t\rho^2(1 + \frac{\alpha_t L}{2}) + \frac{L\sigma^2}{2}\sum_{t=1}^{T}\beta_t^2, \quad (29)$$

since $\mathbb{E}_{\xi^{(N)}}\langle\nabla\mathcal{L}^{meta}(\Theta^{(t)}),\xi^{(t)}\rangle = 0$ and $\mathbb{E}[\|\xi^{(t)}\|_2^2] \leq \sigma^2$, where $\sigma^2$ is the variance of $\xi^{(t)}$. Furthermore, we can deduce that

$$\min_t \mathbb{E}[\|\nabla\mathcal{L}^{meta}(\hat{\mathbf{w}}^{(t)}(\Theta^{(t)}))\|_2^2] \leq \frac{\sum_{t=1}^T(\beta_t - \frac{L\beta_t^2}{2})\mathbb{E}_{\xi^{(N)}}\|\nabla\mathcal{L}^{meta}(\hat{\mathbf{w}}^{(t)}(\Theta^{(t)}))\|_2^2}{\sum_{t=1}^T(\beta_t - \frac{L\beta_t^2}{2})}$$

$$\leq \frac{1}{\sum_{t=1}^T(2\beta_t - L\beta_t^2)}\left[2\mathcal{L}^{meta}(\hat{\mathbf{w}}^{(1)}(\Theta^{(1)})) + \sum_{t=1}^T\alpha_t\rho^2(2+\alpha_t L) + L\sigma^2\sum_{t=1}^T\beta_t^2\right]$$

$$\leq \frac{1}{\sum_{t=1}^T 2\beta_t}\left[2\mathcal{L}^{meta}(\hat{\mathbf{w}}^{(1)}(\Theta^{(1)})) + \sum_{t=1}^T\alpha_t\rho^2(2+\alpha_t L) + L\sigma^2\sum_{t=1}^T\beta_t^2\right]$$

$$\leq \frac{1}{2T\beta_1}\left[2\mathcal{L}^{meta}(\hat{\mathbf{w}}^{(1)}(\Theta^{(1)})) + \alpha_1\rho^2 T(2+L) + L\sigma^2 T\beta_1^2\right]$$

$$= \frac{\mathcal{L}^{meta}(\hat{\mathbf{w}}^{(1)}(\Theta^{(1)}))}{T}\frac{1}{\beta_1} + \frac{\alpha_1\rho^2(2+L)}{\beta_1} + \frac{LT\sigma^2}{2}\beta_1$$

$$= \frac{\mathcal{L}^{meta}(\hat{\mathbf{w}}^{(1)}(\Theta^{(1)}))}{T}\max\{L,\frac{\sigma\sqrt{T}}{c}\} + \min\{1,\frac{k}{T}\}\max\{L,\frac{\sigma\sqrt{T}}{c}\}\rho^2(2+L) + L\sigma^2\frac{c}{\sigma\sqrt{T}}$$

$$\leq \frac{\sigma\mathcal{L}^{meta}(\hat{\mathbf{w}}^{(1)}(\Theta^{(1)}))}{c\sqrt{T}} + \frac{k\sigma\rho^2(2+L)}{c\sqrt{T}} + \frac{L\sigma c}{\sqrt{T}} = \mathcal{O}(\frac{1}{\sqrt{T}}). \tag{30}$$

Therefore, we can conclude that our algorithm can always achieve $\min_{0\leq t\leq T}\mathbb{E}[\|\nabla\mathcal{L}^{meta}(\Theta^{(t)})\|_2^2] \leq \mathcal{O}(\frac{1}{\sqrt{T}})$ in $T$ steps, and this finishes our proof of Theorem 1. $\square$

**Lemma 2.** *(Lemma A.5 in [69]) Let $(a_n)_{n\leq 1}, (b_n)_{n\leq 1}$ be two non-negative real sequences such that the series $\sum_{i=1}^\infty a_n$ diverges, the series $\sum_{i=1}^\infty a_n b_n$ converges, and there exists $K > 0$ such that $|b_{n+1} - b_n| \leq Ka_n$. Then the sequences $(b_n)_{n\leq 1}$ converges to 0.*

**Theorem 2.** *Suppose the loss function $\ell$ is Lipschitz smooth with constant L, and $\mathcal{V}(\cdot)$ is differential with a $\delta$-bounded gradient and twice differential with its Hessian bounded by $\mathcal{B}$, and the loss function $\ell$ have $\rho$-bounded gradients with respect to training/meta data. Let the learning rate $\alpha_t$ satisfies $\alpha_t = \min\{1,\frac{k}{T}\}$, for some $k > 0$, such that $\frac{k}{T} < 1$, and $\beta_t, 1 \leq t \leq N$ is a monotone descent sequence, $\beta_1 = \min\{\frac{1}{L},\frac{c}{\sigma\sqrt{T}}\}$ for some $c > 0$, such that $\frac{\sigma\sqrt{T}}{c} \geq L$ and $\sum_{t=1}^\infty \beta_t \leq \infty, \sum_{t=1}^\infty \beta_t^2 \leq \infty$. Then*

$$\lim_{t\to\infty}\mathbb{E}[\|\nabla\mathcal{L}^{train}(\mathbf{w}^{(t)};\Theta^{(t+1)})\|_2^2] = 0. \tag{31}$$

*Proof.* It is easy to conclude that $\alpha_t$ satisfy $\sum_{t=0}^\infty \alpha_t = \infty, \sum_{t=0}^\infty \alpha_t^2 < \infty$. Recall the update of $\mathbf{w}$ in each iteration as follows:

$$\mathbf{w}^{(t+1)} = \mathbf{w}^{(t)} - \alpha\frac{1}{n}\sum_{i=1}^n \mathcal{V}(L_i^{train}(\mathbf{w}^{(t)});\Theta^{(t+1)})\nabla_{\mathbf{w}}L_i^{train}(\mathbf{w})\Big|_{\mathbf{w}^{(t)}}. \tag{32}$$

It can be written as:

$$\mathbf{w}^{(t+1)} = \mathbf{w}^{(t)} - \alpha_t\nabla\mathcal{L}^{train}(\mathbf{w}^{(t)};\Theta^{(t+1)})|_{\Psi_t}, \tag{33}$$

where $\nabla\mathcal{L}^{train}(\mathbf{w}^{(t)};\Theta) = \frac{1}{n}\sum_{i=1}^n \mathcal{V}(L_i^{train}(\mathbf{w}^{(t)});\Theta)\nabla_{\mathbf{w}}L_i^{train}(\mathbf{w})\Big|_{\mathbf{w}^{(t)}}$. Since the mini-batch $\Psi_t$ is drawn uniformly at random, we can rewrite the update equation as:

$$\mathbf{w}^{(t+1)} = \mathbf{w}^{(t)} - \alpha_t[\nabla\mathcal{L}^{train}(\mathbf{w}^{(t)};\Theta^{(t+1)}) + \psi^{(t)}], \tag{34}$$

where $\psi^{(t)} = \nabla\mathcal{L}^{train}(\mathbf{w}^{(t)};\Theta^{(t+1)})|_{\Psi_t} - \nabla\mathcal{L}^{train}(\mathbf{w}^{(t)};\Theta^{(t+1)})$. Note that $\psi^{(t)}$ is i.i.d. random variable with finite variance, since $\Psi_t$ are drawn i.i.d. with a finite number of samples. Furthermore, $\mathbb{E}[\psi^{(t)}] = 0$, since samples are drawn uniformly at random, and $\mathbb{E}[\|\psi^{(t)}\|_2^2] \leq \sigma^2$.

The objective function $\mathcal{L}^{train}(\mathbf{w}; \Theta)$ defined in Eq. 14 can be easily checked to be Lipschitz-smooth with constant $L$, and have $\rho$-bounded gradients with respect to training data. Observe that

$$\mathcal{L}^{train}(\mathbf{w}^{(t+1)}; \Theta^{(t+2)}) - \mathcal{L}^{train}(\mathbf{w}^{(t)}; \Theta^{(t+1)})$$
$$= \left\{ \mathcal{L}^{train}(\mathbf{w}^{(t+1)}; \Theta^{(t+2)}) - \mathcal{L}^{train}(\mathbf{w}^{(t+1)}; \Theta^{(t+1)}) \right\} + \left\{ \mathcal{L}^{train}(\mathbf{w}^{(t+1)}; \Theta^{(t+1)}) - \mathcal{L}^{train}(\mathbf{w}^{(t)}; \Theta^{(t+1)}) \right\}. \quad (35)$$

For the first term,

$$\mathcal{L}^{train}(\mathbf{w}^{(t+1)}; \Theta^{(t+2)}) - \mathcal{L}^{train}(\mathbf{w}^{(t+1)}; \Theta^{(t+1)})$$
$$= \frac{1}{n} \sum_{i=1}^{n} \left\{ \mathcal{V}(\mathcal{L}_i^{train}(\mathbf{w}^{(t)}); \Theta^{(t+2)}) - \mathcal{V}(\mathcal{L}_i^{train}(\mathbf{w}^{(t)}); \Theta^{(t+1)}) \right\} \mathcal{L}_i^{train}(\mathbf{w}^{(t)})$$
$$\leq \frac{1}{n} \sum_{i=1}^{n} \left\{ \langle \frac{\partial \mathcal{V}(\mathcal{L}_i^{train}(\mathbf{w}^{(t)}); \Theta)}{\partial \Theta} \Big|_{\Theta^{(t)}}, \Theta^{(t+1)} - \Theta^{(t)} \rangle + \frac{\delta}{2} \|\Theta^{(t+1)} - \Theta^{(t)}\|_2^2 \right\} \mathcal{L}_i^{train}(\mathbf{w}^{(t)})$$
$$= \frac{1}{n} \sum_{i=1}^{n} \left\{ \langle \frac{\partial \mathcal{V}(\mathcal{L}_i^{train}(\mathbf{w}^{(t)}); \Theta)}{\partial \Theta} \Big|_{\Theta^{(t)}}, -\beta_t [\nabla \mathcal{L}^{meta}(\hat{\mathbf{w}}^{(t)}(\Theta^{(t)})) + \xi^{(t)}] \rangle + \frac{\delta \beta_t^2}{2} \|\nabla \mathcal{L}^{meta}(\hat{\mathbf{w}}^{(t)}(\Theta^{(t)})) + \xi^{(t)}\|_2^2 \right\} \mathcal{L}_i^{train}(\mathbf{w}^{(t)})$$
$$= \frac{1}{n} \sum_{i=1}^{n} \left\{ \langle \frac{\partial \mathcal{V}(\mathcal{L}_i^{train}(\mathbf{w}^{(t)}); \Theta)}{\partial \Theta} \Big|_{\Theta^{(t)}}, -\beta_t [\nabla \mathcal{L}^{meta}(\hat{\mathbf{w}}^{(t)}(\Theta^{(t)})) + \xi^{(t)}] \rangle + \frac{\delta \beta_t^2}{2} \|\nabla \mathcal{L}^{meta}(\hat{\mathbf{w}}^{(t)}(\Theta^{(t)})) + \xi^{(t)}\|_2^2 \right\} \mathcal{L}_i^{train}(\mathbf{w}^{(t)})$$
$$= \frac{1}{n} \sum_{i=1}^{n} \left\{ \langle \frac{\partial \mathcal{V}(\mathcal{L}_i^{train}(\mathbf{w}^{(t)}); \Theta)}{\partial \Theta} \Big|_{\Theta^{(t)}}, -\beta_t [\nabla \mathcal{L}^{meta}(\hat{\mathbf{w}}^{(t)}(\Theta^{(t)})) + \xi^{(t)}] \rangle + \frac{\delta \beta_t^2}{2} (\|\nabla \mathcal{L}^{meta}(\hat{\mathbf{w}}^{(t)}(\Theta^{(t)}))\|_2^2 + \|\xi^{(t)}\|_2^2 + \right.$$
$$\left. 2\langle \nabla \mathcal{L}^{meta}(\hat{\mathbf{w}}^{(t)}(\Theta^{(t)})), \xi^{(t)} \rangle) \right\} L_i^{train}(\mathbf{w}^{(t)}) \quad (36)$$

For the second term,

$$\mathcal{L}^{train}(\mathbf{w}^{(t+1)}; \Theta^{(t+1)}) - \mathcal{L}^{train}(\mathbf{w}^{(t)}; \Theta^{(t+1)})$$
$$\leq \langle \nabla \mathcal{L}^{train}(\mathbf{w}^{(t)}; \Theta^{(t+1)}), \mathbf{w}^{(t+1)} - \mathbf{w}^{(t)} \rangle + \frac{L}{2} \|\mathbf{w}^{(t+1)} - \mathbf{w}^{(t)}\|_2^2$$
$$= \langle \nabla \mathcal{L}^{train}(\mathbf{w}^{(t)}; \Theta^{(t+1)}), -\alpha_t [\nabla \mathcal{L}^{train}(\mathbf{w}^{(t)}; \Theta^{(t+1)}) + \psi^{(t)}] \rangle + \frac{L\alpha_t^2}{2} \|\nabla \mathcal{L}^{train}(\mathbf{w}^{(t)}; \Theta^{(t+1)}) + \psi^{(t)}\|_2^2$$
$$= -(\alpha_t - \frac{L\alpha_t^2}{2}) \|\nabla \mathcal{L}^{train}(\mathbf{w}^{(t)}; \Theta^{(t+1)})\|_2^2 + \frac{L\alpha_t^2}{2} \|\psi^{(t)}\|_2^2 - (\alpha_t - L\alpha_t^2) \langle \nabla \mathcal{L}^{train}(\mathbf{w}^{(t)}; \Theta^{(t+1)}), \psi^{(t)} \rangle. \quad (37)$$

Therefore, we have:

$$\mathcal{L}^{train}(\mathbf{w}^{(t+1)}; \Theta^{(t+2)}) - \mathcal{L}^{train}(\mathbf{w}^{(t)}; \Theta^{(t+1)}) \leq \frac{1}{n} \sum_{i=1}^{n} \left\{ \langle \frac{\partial \mathcal{V}(\mathcal{L}_i^{train}(\mathbf{w}^{(t)}); \Theta)}{\partial \Theta} \Big|_{\Theta^{(t)}}, -\beta_t [\nabla \mathcal{L}^{meta}(\hat{\mathbf{w}}^{(t)}(\Theta^{(t)})) + \xi^{(t)}] \rangle + \right.$$
$$\left. + \frac{\delta \beta_t^2}{2} (\|\nabla \mathcal{L}^{meta}(\hat{\mathbf{w}}^{(t)}(\Theta^{(t)}))\|_2^2 + \|\xi^{(t)}\|_2^2 + 2\langle \nabla \mathcal{L}^{meta}(\hat{\mathbf{w}}^{(t)}(\Theta^{(t)})), \xi^{(t)} \rangle) \right\} L_i^{train}(\mathbf{w}^{(t)})$$
$$- (\alpha_t - \frac{L\alpha_t^2}{2}) \|\nabla \mathcal{L}^{train}(\mathbf{w}^{(t)}; \Theta^{(t+1)})\|_2^2 + \frac{L\alpha_t^2}{2} \|\psi^{(t)}\|_2^2 - (\alpha_t - L\alpha_t^2) \langle \nabla \mathcal{L}^{train}(\mathbf{w}^{(t)}; \Theta^{(t+1)}), \psi^{(t)} \rangle. \quad (38)$$

Taking expectation of both sides of (37) and since $\mathbb{E}[\xi^{(t)}] = 0, \mathbb{E}[\psi^{(t)}] = 0$, we have

$$\mathbb{E}[\mathcal{L}^{train}(\mathbf{w}^{(t+1)}; \Theta^{(t+2)})] - \mathbb{E}[\mathcal{L}^{train}(\mathbf{w}^{(t)}; \Theta^{(t+1)})] \leq \mathbb{E} \frac{1}{n} \sum_{i=1}^{n} L_i^{train}(\mathbf{w}^{(t)}) \left\{ \langle \frac{\partial \mathcal{V}(\mathcal{L}_i^{train}(\mathbf{w}^{(t)}); \Theta)}{\partial \Theta} \Big|_{\Theta^{(t)}}, -\beta_t [\nabla \mathcal{L}^{meta}(\hat{\mathbf{w}}^{(t)}(\Theta^{(t)}))] \rangle + \right.$$
$$\left. + \frac{\delta \beta_t^2}{2} (\|\nabla \mathcal{L}^{meta}(\hat{\mathbf{w}}^{(t)}(\Theta^{(t)}))\|_2^2 + \|\xi^{(t)}\|_2^2) \right\} - \alpha_t \mathbb{E}[\|\nabla \mathcal{L}^{train}(\mathbf{w}^{(t)}; \Theta^{(t+1)})\|_2^2] + \frac{L\alpha_t^2}{2} \left\{ \mathbb{E}[\|\nabla \mathcal{L}^{train}(\mathbf{w}^{(t)}; \Theta^{(t+1)})\|_2^2] + \mathbb{E}[\|\psi^{(t)}\|_2^2] \right\}$$

Summing up the above inequalities over $t = 1, ..., \infty$ in both sides, we obtain

$$\sum_{t=1}^{\infty} \alpha_t \mathbb{E}[\|\nabla \mathcal{L}^{train}(\mathbf{w}^{(t)}; \Theta^{(t+1)})\|_2^2] + \sum_{t=1}^{\infty} \beta_t \mathbb{E} \frac{1}{n} \sum_{i=1}^{n} \|L_i^{train}(\mathbf{w}^{(t)})\| \|\frac{\partial \mathcal{V}(\mathcal{L}_i^{train}(\mathbf{w}^{(t)}); \Theta)}{\partial \Theta} \Big|_{\Theta^{(t)}}\| \cdot \|\nabla \mathcal{L}^{meta}(\hat{\mathbf{w}}^{(t)}(\Theta^{(t)}))\|$$
$$\leq \sum_{t=1}^{\infty} \frac{L\alpha_t^2}{2} \left\{ \mathbb{E}[\|\nabla \mathcal{L}^{train}(\mathbf{w}^{(t)}; \Theta^{(t+1)})\|_2^2] + \mathbb{E}[\|\psi^{(t)}\|_2^2] \right\} + \mathbb{E}[\mathcal{L}^{train}(\mathbf{w}^{(1)}; \Theta^{(2)})] - \lim_{T \to \infty} \mathbb{E}[\mathcal{L}^{train}(\mathbf{w}^{(t+1)}; \Theta^{(t+2)})]$$
$$+ \sum_{t=1}^{\infty} \frac{\delta \beta_t^2}{2} \left\{ \frac{1}{n} \sum_{i=1}^{n} \|L_i^{train}(\mathbf{w}^{(t)})\| (\mathbb{E}\|\nabla \mathcal{L}^{meta}(\hat{\mathbf{w}}^{(t)}(\Theta^{(t)}))\|_2^2 + \mathbb{E}\|\xi^{(t)}\|_2^2 \right\}$$
$$\leq \sum_{t=1}^{\infty} \frac{L\alpha_t^2}{2} \{\rho^2 + \sigma^2\} + \mathbb{E}[\mathcal{L}^{tr}(\mathbf{w}^{(1)}; \Theta^{(2)})] + \sum_{t=1}^{\infty} \frac{\delta \beta_t^2}{2} \{M(\rho^2 + \sigma^2)\} \leq \infty.$$

The last inequality holds since $\sum_{t=0}^{\infty} \alpha_t^2 < \infty$, $\sum_{t=0}^{\infty} \beta_t^2 < \infty$, and $\frac{1}{n} \sum_{i=1}^{n} \|L_i^{train}(\mathbf{w}^{(t)})\| \leq M$ for limited number of samples' loss is bounded. Thus we have

$$\sum_{t=1}^{\infty} \alpha_t \mathbb{E}[\|\nabla \mathcal{L}^{train}(\mathbf{w}^{(t)}; \Theta^{(t+1)})\|_2^2] + \sum_{t=1}^{\infty} \beta_t \mathbb{E} \frac{1}{n} \sum_{i=1}^{n} \|L_i^{train}(\mathbf{w}^{(t)})\| \|\frac{\partial \mathcal{V}(\mathcal{L}_i^{train}(\mathbf{w}^{(t)}); \Theta)}{\partial \Theta} \Big|_{\Theta^{(t)}}\| \cdot \|\nabla \mathcal{L}^{meta}(\hat{\mathbf{w}}^{(t)}(\Theta^{(t)}))\| \leq \infty.$$
$$(39)$$

Since

$$\sum_{t=1}^{\infty} \beta_t \mathbb{E} \frac{1}{n} \sum_{i=1}^{n} \|L_i^{train}(\mathbf{w}^{(t)})\| \|\frac{\partial \mathcal{V}(\mathcal{L}_i^{train}(\mathbf{w}^{(t)});\Theta)}{\partial \Theta}\Big|_{\Theta^{(t)}}\| \cdot \|\nabla \mathcal{L}^{meta}(\hat{\mathbf{w}}^{(t)}(\Theta^{(t)}))\|$$

$$\leq M\delta\rho \sum_{t=1}^{\infty} \beta_t \leq \infty, \tag{40}$$

which implies that $\sum_{t=1}^{\infty} \alpha_t \mathbb{E}[\|\nabla \mathcal{L}^{train}(\mathbf{w}^{(t)};\Theta^{(t+1)})\|_2^2] < \infty$. By Lemma 2, to substantiate $\lim_{t\to\infty} \mathbb{E}[\nabla \mathcal{L}^{train}(\mathbf{w}^{(t)};\Theta^{(t+1)})\|_2^2] = 0$, since $\sum_{t=0}^{\infty} \alpha_t = \infty$, it only needs to prove:

$$\left|\mathbb{E}[\nabla \mathcal{L}^{train}(\mathbf{w}^{(t+1)};\Theta^{(t+2)})\|_2^2] - \mathbb{E}[\nabla \mathcal{L}^{train}(\mathbf{w}^{(t)};\Theta^{(t+1)})\|_2^2]\right| \leq C\alpha_k, \tag{41}$$

for some constant $C$. Based on the inequality:

$$|(\|a\| + \|b\|)(\|a\| - \|b\|)| \leq \|a + b\|\|a - b\|, \tag{42}$$

we then have:

$$\left|\mathbb{E}[\|\nabla \mathcal{L}^{train}(\mathbf{w}^{(t+1)};\Theta^{(t+2)})\|_2^2] - \mathbb{E}[\|\nabla \mathcal{L}^{train}(\mathbf{w}^{(t)};\Theta^{(t+1)})\|_2^2]\right|$$

$$= \left|\mathbb{E}\left[(\|\nabla \mathcal{L}^{train}(\mathbf{w}^{(t+1)};\Theta^{(t+2)})\|_2 + \|\nabla \mathcal{L}^{train}(\mathbf{w}^{(t)};\Theta^{(t+1)})\|_2)(\|\nabla \mathcal{L}^{train}(\mathbf{w}^{(t+1)};\Theta^{(t+2)})\|_2 - \|\nabla \mathcal{L}^{train}(\mathbf{w}^{(t)};\Theta^{(t+1)})\|_2)\right]\right|$$

$$\leq \mathbb{E}\left[\left|\|\nabla \mathcal{L}^{train}(\mathbf{w}^{(t+1)};\Theta^{(t+1)})\|_2 + \|\nabla \mathcal{L}^{train}(\mathbf{w}^{(t)};\Theta^{(t)})\|_2\right|\left|(\|\nabla \mathcal{L}^{train}(\mathbf{w}^{(t+1)};\Theta^{(t+2)})\|_2 - \|\nabla \mathcal{L}^{train}(\mathbf{w}^{(t)};\Theta^{(t+1)})\|_2)\right|\right]$$

$$\leq \mathbb{E}\left[\left\|\nabla \mathcal{L}^{train}(\mathbf{w}^{(t+1)};\Theta^{(t+2)}) + \nabla \mathcal{L}^{train}(\mathbf{w}^{(t)};\Theta^{(t+1)})\right\|_2 \left\|\nabla \mathcal{L}^{train}(\mathbf{w}^{(t+1)};\Theta^{(t+2)}) - \nabla \mathcal{L}^{train}(\mathbf{w}^{(t)};\Theta^{(t+1)})\right\|_2\right]$$

$$\leq \mathbb{E}\left[(\left\|\nabla \mathcal{L}^{train}(\mathbf{w}^{(t+1)};\Theta^{(t+2)})\right\|_2 + \left\|\nabla \mathcal{L}^{train}(\mathbf{w}^{(t)};\Theta^{(t+1)})\right\|_2) \left\|\nabla \mathcal{L}^{train}(\mathbf{w}^{(t+1)};\Theta^{(t+2)}) - \nabla \mathcal{L}^{train}(\mathbf{w}^{(t)};\Theta^{(t+1)})\right\|_2\right]$$

$$\leq 2L\rho \mathbb{E}\left[\|(\mathbf{w}^{(t+1)},\Theta^{(t+2)}) - (\mathbf{w}^{(t)},\Theta^{(t+1)})\|_2\right]$$

$$\leq 2L\rho\alpha_t\beta_t \mathbb{E}\left[\left\|\left(\nabla \mathcal{L}^{train}(\mathbf{w}^{(t)};\Theta^{(t+1)}) + \psi^{(t)}, \nabla \mathcal{L}^{meta}(\Theta^{(t+1)}) + \xi^{(t+1)}\right)\right\|_2\right]$$

$$\leq 2L\rho\alpha_t\beta_t \mathbb{E}\left[\sqrt{\|\nabla \mathcal{L}^{train}(\mathbf{w}^{(t)};\Theta^{(t+1)}) + \psi^{(t)}\|_2^2} + \sqrt{\|\nabla \mathcal{L}^{meta}(\Theta^{(t+1)}) + \xi^{(t+1)}\|_2^2}\right]$$

$$\leq 2L\rho\alpha_t\beta_t \sqrt{\mathbb{E}\left[\|\nabla \mathcal{L}^{train}(\mathbf{w}^{(t)};\Theta^{(t+1)}) + \psi^{(t)}\|_2^2\right] + \mathbb{E}\left[\|\nabla \mathcal{L}^{meta}(\Theta^{(t+1)}) + \xi^{(t+1)}\|_2^2\right]}$$

$$\leq 2L\rho\alpha_t\beta_t \sqrt{\mathbb{E}\left[\|\nabla \mathcal{L}^{train}(\mathbf{w}^{(t)};\Theta^{(t+1)})\|_2^2\right] + \mathbb{E}\left[\|\psi^{(t)}\|_2^2\right] + \mathbb{E}\left[\|\xi^{(t+1)}\|_2^2\right] + \mathbb{E}\left[\|\nabla \mathcal{L}^{meta}(\Theta^{(t+1)})\|_2^2\right]}$$

$$\leq 2L\rho\alpha_t\beta_t \sqrt{2\sigma^2 + 2\rho^2}$$

$$\leq 2\sqrt{2(\sigma^2 + \rho^2)} L\rho\beta_1\alpha_t. \tag{43}$$

According to the above inequality, we can conclude that our algorithm can achieve

$$\lim_{t\to\infty} \mathbb{E}[\|\nabla \mathcal{L}^{train}(\mathbf{w}^{(t)};\Theta^{(t+1)})\|_2^2] = 0. \tag{44}$$

The proof is completed. $\qquad\square$

# D Experimentalp Details on Meta-Weight-Net

To match the original training step size, in our experiment, we can consider normalizing the weights of all examples in a training batch so that they sum up to one. In other words, we choose to have a hard constraint within the set $\|\mathcal{V}(\ell;\Theta)\|_1 = 1$, and the normalized weight

$$\eta_i^{(t)} = \frac{\mathcal{V}^{(t)}(L_i;\Theta)}{\sum_j \mathcal{V}^{(t)}(L_j;\Theta) + \delta(\sum_i \mathcal{V}^{(t)}(L_j;\Theta))}, \tag{45}$$

where the function of $\delta(\cdot)$ is to prevent the degeneration case when all $\mathcal{V}^{(t)}(L_j;\Theta)$ in a mini-batch are zeros, i.e. $\delta(a) = \tau$. $\tau$ denotes a constant grater than 0, if $a = 0$; and equal to 0 otherwise. With batch normalization, we effectively cancel the learning rate of Meta-Weight-Net, and it works well with a fixed learning rate.

# E MLP architecture of Meta-Weight-Net

We actually have tried different MLP architecture settings in experiments. The right table depicts some representative results under 6 different structures, with different depths and widths. It can be seen that varying MLP settings have unsubstantial effects to the final result. We thus prefer to use the simple and shallow one.

Table 5: Test accuracy on CIFAR-10 and CIFAR-100 of different MW-Nets.

| architcture | Imbalance (factor 100) | | Uniform noise (40%) | | Flip noise (40%) | |
|---|---|---|---|---|---|---|
| | CIFAR10 | CIFAR100 | CIFAR10 | CIFAR100 | CIFAR10 | CIFAR100 |
| 1-50-1 | 73.50 | 41.87 | 89.01 | 67.63 | 87.38 | 57.83 |
| 1-100-1 | 75.21 | 42.09 | 89.27 | 67.73 | 87.54 | 58.64 |
| 1-200-1 | 74.70 | 41.72 | 89.58 | 67.84 | 87.74 | 58.41 |
| 1-100-100-1 | 75.01 | 41.97 | 89.09 | 66.48 | 87.28 | 57.39 |
| 1-10-10-1 | 74.71 | 41.94 | 89.10 | 66.53 | 87.58 | 57.11 |
| 1-10-10-10-1 | 74.96 | 42.31 | 88.82 | 66.67 | 87.36 | 57.29 |

(a) CIFAR-10 40% noise

(b) CIFAR-10 60% noise

(c) CIFAR-100 40% noise

(d) CIFAR-100 60% noise

Figure 7: Training and test accuracy changing curves in **uniform noise** cases of CIFAR-10 and CIFAR-100 datasets. Solid and dotted curves denote the test and training accuracies, respectively. Our method and L2RW are less prone to overfit label noises, while our method can converge faster at around 40 epoch as shown in Fig. 7(a). We thus terminate our method in 40 epoch in other experiments.

# F    Complexity Analysis of The Proposed Algorithm

Our Meta-PGC algorithm can be roughly regarded as that requires an extra full forward and backward passes of the network on training data (step 5 in algorithm) and an extra full forward and backward passes of the network of meta data (step 6 in algorithm) in the presence of the normal classifier network' parameters update (step 7 in algorithm). Therefore compared to regular training, our method needs approximately $3\times$ training time in each iteration. It is suggest to let batch size of meta data less than or equal to batch size of training data, which avoids GPU memory increase and speeds up the learning process.

# G    Robustness Towards Label Noise Overfitting Issue

Fig. 7 plots the tendency curves of the mini-batch training accuracy calculated on noisy training set in experiments, as well as those calculated simultaneously on clean test data during learning iterations. From the figure, we can easily find that the BaseModel can easily overfit to the noisy labels contained in the training set, whose test accuracy quickly degrades after the first learning rate decays. While our method and L2RW are less prone to such overfitting issue, and they retain the similar test accuracy until termination. Especially, throughout all our experiments, we find that our method can converge

significantly faster than the BaseModel and L2RW methods[1], as clearly shown in Fig.7, and get the peak performance at around 40 epochs, as compared with 120 epochs required for the other two methods, as shown in Fig. 7(a). We thus only report our results at 40 epochs in the figure for other experiments.

# H    Confusion Matrices for Class Imbalance and Corrupted Labels

We demonstrate confusion matrices of Baseline and our algorithm on long-tailed CIFAR-10 dataset for class imbalance and corrupted labels experiments, as shown in Fig. 2-4.

Figure 8: Confusion matrices on long-tailed CIFAR-10 with imbalance factors ranging from 1 to 200.

Figure 9: Confusion matrices on CIFAR-10 dataset with varying noise rates under **uniform noise**.

Figure 10: Confusion matrices on CIFAR-10 dataset with varying noise rates under **flip noise**.

# I Convergence Verification for The Training loss and Meta Loss

To validate the convergence results obtained in Theorem 1 and 2 in the paper, we plot the changing tendency curves of training and meta losses with the number of epochs in our experiments, as shown in Fig. 11 - 13. The convergence tendency can be easily observed in the figures, substantiating the properness of the theoretical results in two theorems.

| (a) 1 for CIFAR-10 | (b) 10 for CIFAR-10 | (c) 20 for CIFAR-10 | (d) 50 for CIFAR-10 |

| (e) 100 for CIFAR-10 | (f) 200 for CIFAR-10 | (g) 1 for CIFAR-100 | (h) 10 for CIFAR-100 |

| (i) 20 for CIFAR-100 | (j) 50 for CIFAR-100 | (k) 100 for CIFAR-100 | (l) 200 for CIFAR-100 |

Figure 11: Training and meta loss tendency curves on long-tailed CIFAR with imbalance factors ranging from 1 to 200.

| (a) CIFAR-10 40% | (b) CIFAR-10 60% | (c) CIFAR-100 40% | (d) CIFAR-100 60% |

Figure 12: Training and meta loss tendency curves on CIFAR dataset with varying noise rates under **uniform noise**.

| (a) CIFAR-10 20% | (b) CIFAR-10 40% | (c) CIFAR-100 20% | (d) CIFAR-100 40% |

Figure 13: Training and meta loss tendency curves on CIFAR dataset with varying noise rates under **flip noise**.