[Reviews · NeurIPS 2019]

Reviewer 1



The paper presents a novel meta-learning method as well as the detailed algorithm and analysis of its convergence property. The proposed method can adaptively learn an explicit weighting function directly from data. The weighting function is a MLP with one hidden layer and the method can fit a wide range of weighting functions. The paper discusses lots of related work and analyzes the pros and cons. Experimental results show that the proposed method achieves better performance compared with the state-of-the-art methods. The paper is well organized, the figures and tables are clear. But there are also some handwriting errors. For example, at line 180, there should be “stable” rather than “stale”. At line 69, there should be "tradition" instead of "traditional".

Reviewer 2



This paper proposed to reweight samples using a simple one-layer MLP in a meta-learning manner. The proposed method is both theoretically and empirically justified. Theoretically, the convergence of the proposed method is proofed. Empirically, the proposed method is justified in both class imbalance and noisy label problems. Learning sample-reweighting from data is not a new thing. As introduced in related work section, there are other methods in this line, e.g. MentorNet and Learning to Reweight (L2RW). MentorNet also learns to reweight samples from data with the aid of a neural network (a LSTM). Can you discuss more about how the choice of these explicit reweighting functions influence the results? In noisy label experiments, the classifiers are trained in only 40 epochs for uniform noise, and 60 epochs for flip noise. My concern is that not all methods can converge in too few epochs. It more be more clearly if the whole tendencies of different methods are compared with more training epochs, e.g. 200 epochs. Without seeing the whole tendencies, we cannot simply say that the proposed method is converged faster than other methods (as claimed in line 101 of supplementary materials).

Reviewer 3



This paper studies the problem of learning from biased training data (i.e. distribution shift) with the help of a small set of unbiased meta-data. This covers notably the case of class imbalance and noisy label. The proposed meta-weight-net is an MLP with one hidden layer that learns a mapping from training loss of a sample to its weight. Minimizing the training objectives naturally leads us to focus more on samples that agree with the meat-knowledge. Theoretically it is shown that the algorithm converges to critical points of the loss under classical assumptions (but I am quite confused by the proof, see below). The experimental results are very promising. - Pros: The problem that is studied by this paper is important and the idea of learning a meta-weight-net is interesting and reasonable. On the one hand, we get rid of the burden of weight function design and hyperparameter tuning compared with adhoc sample reweighting strategies. Although this comes at the cost of having some meta data, such requirement should be feasible in many cases. On the other hand, compared with other weight learning methods, here one only learns a mapping from training loss to sample weight. The procedure is thus simpler but should be sufficient if we suppose there is some regularity in the optimal weight that needs to be assigned to each sample and this weight is related to the training loss of the sample. The experimental part is very complete. In both class imbalance and noisy label settings, the authors compare with a bunch of baseline methods and show that meta-weight-net learning effectively performs the best most of the time. Different datasets (though all of them are image ones) and model structures are considered, strengthening the above claim. The experiments are also conducted with different level of class imbalance or label noise. - Questions/issues: My biggest concern is on the correctness of the theorems and proofs, though it constitutes a less important part of this paper. In fact, I can not understand how \mathcal{L}^{meta}(\Theta) is defined in Theorem 1 and appendix. If we use (2) of the main paper as the definition, it seems that we need somehow a way to compute w*. By the way, the notation suggests rather \mathcal{L}^{meta} is a function of w, and the same holds for (4) of the supplementary material. On the other hand, if we use line 33 of the supplementary, which seems to be the case in the proof, I do not understand how it is possible that \mathcal{L}^{meta} does not depend on w. Meanwhile, suppose that there is a dependence of \mathcal{L}^{meta} on w, the proof of theorem 1 cannot hold anymore. For example, for the inequality after line 47 (supp), \Theta^(t+1) and \Theta^(t) are not evaluated with the same w so I have a doubt on whether we can write an inequality like this. In lemma 1, it seems that it is proved that the gradient of \mathcal{L}^{meta} is bounded (with the definition given in line 33-supp). Nonetheless, to prove that \mathcal{L}^{meta} is Lipshitz smooth we need to prove that the Jacobian of the gradient operator is bounded. Finally, in the proof of theorem 2 line 71, the authors claim \sum... < \infty. I can not see how we can draw such a conclusion since we do not really have a telescopic sum in (21) (with two \Theta^{t+1} at the left of the inequality). To conclude, I feel the proof should be much more involved to prove some kind of convergence results, and I have a doubt on the current proofs of the two theorems. - Minor points: 1. In the mathematical formula after line 75 in supplementary material, shouldn't there be a l2 norm in the sixth line and a different bound for \mathcal{L}^{meta} instead of \rho in the end? There seem to be some other typos but I would like to first understand the points that I mention above so I do not list them here. 2. I do not like the fact the paper puts emphasis on that a neural network with one hidden layer is a universal approximator. This is just a theoretical result and gives little insights into the true capacity of a neural network with a fixed number of neurons. I think the important thing here is the optimal weighting function may be simple enough to be approximated by an MLP with a single hidden layer. ---- At the beginning I tend towards voting for accepting the paper since the proposed algorithm is sensible and the experimental part is strong. However, there might be some misunderstanding in my understanding of the theorems and their proofs. As I would like to avoid accepting papers with wrong theoretical statements (even though these might not be that important for this paper), I cannot vote for accept for the time being. ============================================================== After the rebuttal I would like to thank the authors for their detailed reply. The authors' rebuttal has clarified my concern on the definition of \mathcal{L}^{meta}. However, I still have some doubts on how the telescopic sums are formulated in the proof of Theorem 1 and 2. For example in the equation following line 26 in the rebuttal, there are w^(t+1),\Theta^(t+1) and w^(t),\Theta^(t+1) which will not disappear after taking the sum. The same question also occurs for the equation following line 42. I hope these questions can be resolved after the revision.

[Author Response · NeurIPS 2019]



**To Reviewer #1:**

**Q1.1: MLP architecture of Meta-Weight-Net.** We actually have tried different MLP architecture settings in experiments. The right table depicts some representative results under 6 different structures, with different depths and widths. It can be seen that varying MLP settings have unsubstantial effects to the final result. We thus prefer to use the simple and shallow one.

| architcture | Imbalance (factor 100) | | Uniform noise (40%) | | Flip noise (40%) | |
|---|---|---|---|---|---|---|
| | CIFAR10 | CIFAR100 | CIFAR10 | CIFAR100 | CIFAR10 | CIFAR100 |
| 1-50-1 | 73.50 | 41.87 | 89.01 | 67.63 | 87.38 | 57.83 |
| 1-100-1 | 75.21 | 42.09 | 89.27 | 67.73 | 87.54 | 58.64 |
| 1-200-1 | 74.70 | 41.72 | 89.58 | 67.84 | 87.74 | 58.41 |
| 1-100-100-1 | 75.01 | 41.97 | 89.09 | 66.48 | 87.28 | 57.39 |
| 1-10-10-1 | 74.71 | 41.94 | 89.10 | 66.53 | 87.58 | 57.11 |
| 1-10-10-10-1 | 74.96 | 42.31 | 88.82 | 66.67 | 87.36 | 57.29 |

**To Reviewer #2:**

**Q2.1: How the choices of weight function influence the results?** Succeeded from the understanding of conventional sample reweighting approaches, this explicit weighting function is set as mapping from loss to weight, and thus MLP is suitable. Instead, since LSTM is functioned on temporal feature input, it is not proper to be used here. As introduced in Q1.1, we have also tested different structures for MW-Net (with different depths and widths), which have only unsubstantial influence to the final result.

**Q2.2: Experiment results with more training epochs.** In our experiments, we have tried to specifically set the epoch number for each compared method to guarantee possibly the optimal performance. Actually, we have shown in Fig. 1(a) of SM the performance tendency of our method with more than 100 epochs. It is easy to see the convergence of our method after about 40 epochs. Similar phenomena have been observed from all our experiments. Comparatively, most of other methods could get the best performance before 100 epochs, while the state-of-the-art L2RW needs more than 100 epochs, as shown in Fig. 1(a) of SM as well as Fig. 6 of the paper. This supports us to say that our method converges relatively faster. We'll add more results in revision for more clarification.

**To Reviewer #3:**

**Q3.1: Definition of $\mathcal{L}^{meta}(\Theta)$ and the proof of the inequality after line 47 (supp).** We sincerely thank the reviewer for pointing this out. The function $\mathcal{L}^{(meta)}$ does depend on $\mathbf{w}$, and thus it should be inappropriate to neglect the symbol $\mathbf{w}$. Specifically, in our algorithm (Algorithm 1), we use one step gradient descend result $\hat{\mathbf{w}}^{(t)}(\Theta)$ as the variable in $\mathcal{L}^{meta}$ function, and $\Theta^{(t+1)}$ and $\Theta^{(t)}$ appearing between Line 47-48 of SM do be evaluated with different $\mathbf{w}$s, which should be under $\hat{\mathbf{w}}^{(t+1)}$ and $\hat{\mathbf{w}}^{(t)}$, respectively, just as the reviewer properly indicates. The deduction under line 47 thus should be rectified as follows:

$$\mathcal{L}^{meta}(\hat{\mathbf{w}}^{(t+1)}(\Theta^{(t+1)})) - \mathcal{L}^{meta}(\hat{\mathbf{w}}^{(t)}(\Theta^{(t)})) = \{\mathcal{L}^{meta}(\hat{\mathbf{w}}^{(t+1)}(\Theta^{(t+1)})) - \mathcal{L}^{meta}(\hat{\mathbf{w}}^{(t)}(\Theta^{(t+1)}))\} + \{\mathcal{L}^{meta}(\hat{\mathbf{w}}^{(t)}(\Theta^{(t+1)})) - \mathcal{L}^{meta}(\hat{\mathbf{w}}^{(t)}(\Theta^{(t)}))\}.$$

The deduction below line 47 in SM actually deduces the upper bound of the above second term (difference of $\mathcal{L}^{(meta)}$ under the same $w^{(t)}$). That is, let $Re = -(\beta_t - \frac{L\beta_t^2}{2})\|\nabla\mathcal{L}^{meta}(\hat{\mathbf{w}}^{(t)}(\Theta^{(t)}))\|_2^2 + \frac{L\beta_t^2}{2}\|\xi^{(t)}\|_2^2 - (\beta_t - L\beta_t^2)\langle\nabla\mathcal{L}^{meta}(\hat{\mathbf{w}}^{(t)}(\Theta^{(t)})), \xi^{(t)}\rangle$ we then have

$$\mathcal{L}^{meta}(\hat{\mathbf{w}}^{(t+1)}(\Theta^{(t+1)})) - \mathcal{L}^{meta}(\hat{\mathbf{w}}^{(t)}(\Theta^{(t)})) \leq \mathcal{L}^{meta}(\hat{\mathbf{w}}^{(t+1)}(\Theta^{(t+1)})) - \mathcal{L}^{meta}(\hat{\mathbf{w}}^{(t)}(\Theta^{(t+1)})) + Re.$$

Summing up the above inequalities and rearranging the terms, we can obtain

$$\sum_{t=1}^{T}(\beta_t - \frac{L\beta_t^2}{2})\|\nabla\mathcal{L}^{meta}(\Theta^{(t)})\|_2^2 \leq \mathcal{L}^{meta}(\hat{\mathbf{w}}^{(T+1)}(\Theta^{(T+1)})) - \mathcal{L}^{meta}(\hat{\mathbf{w}}^{(2)}(\Theta^{(1)})) + \mathcal{L}^{meta}(\hat{\mathbf{w}}^{(1)}(\Theta^{(1)})) - \mathcal{L}^{meta}(\hat{\mathbf{w}}^{(T+1)}(\Theta^{(T+1)}))$$

$$-\sum_{t=1}^{T}(\beta_t - L\beta_t^2)\langle\nabla\mathcal{L}^{meta}(\Theta^{(t)}), \xi^{(t)}\rangle + \frac{L}{2}\sum_{t=1}^{T}\beta_t^2\|\xi^{(t)}\|_2^2 = \mathcal{L}^{meta}(\hat{\mathbf{w}}^{(1)}(\Theta^{(1)})) - \mathcal{L}^{meta}(\hat{\mathbf{w}}^{(2)}(\Theta^{(1)})) - \sum_{t=1}^{T}(\beta_t - L\beta_t^2)\langle\nabla\mathcal{L}^{meta}(\Theta^{(t)}), \xi^{(t)}\rangle + \frac{L}{2}\sum_{t=1}^{T}\beta_t^2\|\xi^{(t)}\|_2^2,$$

This is almost similar to (14), with only $\mathcal{L}^{meta}(\Theta^{(1)}) - \mathcal{L}^{meta}(\Theta^*)$ replaced by $\mathcal{L}^{meta}(\hat{\mathbf{w}}^{(1)}(\Theta^{(1)})) - \mathcal{L}^{meta}(\hat{\mathbf{w}}^{(2)}(\Theta^{(1)}))$. So do the following inequalities (15)(16). We'll revise the proof accordingly to avoid possible confusions of readers.

**Q3.2: Prove that $\mathcal{L}^{(meta)}$ is Lipshitz smooth in lemma 1.** Many thanks to the reviewer for carefully checking our proof. To guarantee that Eq.(9) holds based on the proof already being presented, we do need to additionally prove $\nabla_{\Theta^2}^2 L_i^{meta}(\hat{\mathbf{w}}^{(t)}(\Theta))\big|_{\Theta^{(t)}}$ is bounded, which needs another two mild conditions: the meta loss function is Lipschitz smooth with constant $L$, and $\mathcal{V}(\cdot)$ is a twice differential with its Hessian bounded by $\mathcal{B}$. The proof is then presented as follows: Let $v_j(\Theta) = \mathcal{V}(L_j^{train}(\mathbf{w}^{(t)}); \Theta)$ and $G_{ij}$ being defined in line 18 of SM, taking gradient of $\Theta$ in both sides of (6), we have

$$\nabla_{\Theta^2}^2 L_i^{meta}(\hat{\mathbf{w}}^{(t)}(\Theta))\big|_{\Theta^{(t)}} = \frac{-\alpha}{n}\sum_{j=1}^{n}\left[\frac{\partial}{\partial\Theta}(G_{ij})\big|_{\Theta^{(t)}}\frac{\partial v_j(\Theta)}{\partial\Theta}\big|_{\Theta^{(t)}} + (G_{ij})\frac{\partial^2 v_j(\Theta)}{\partial\Theta^2}\big|_{\Theta^{(t)}}\right].$$

For the first term in the right hand side, we have that $\left\|\frac{\partial}{\partial\Theta}(G_{ij})\big|_{\Theta^{(t)}}\frac{\partial v_j(\Theta)}{\partial\Theta}\big|_{\Theta^{(t)}}\right\| \leq \rho\left\|\frac{\partial}{\partial\Theta}\left(\frac{\partial L_i^{meta}(\hat{\mathbf{w}})}{\partial\Theta}\big|_{\Theta^{(t)}}\right)\big|^T\frac{\partial L_j^{train}(\mathbf{w})}{\partial\mathbf{w}}\big|_{\mathbf{w}^{(t)}}\right\|$

$= \rho\left\|\frac{\partial}{\partial\hat{\mathbf{w}}}\left(\frac{\partial L_i^{meta}(\hat{\mathbf{w}})}{\partial\hat{\mathbf{w}}}\big|_{\hat{\mathbf{w}}^{(t)}}\frac{-\alpha}{n}\sum_{j=1}^{n}\frac{\partial L_j^{train}(\mathbf{w})}{\partial\mathbf{w}}\big|_{\mathbf{w}^{(t)}}\frac{\partial v_j(\Theta)}{\partial\Theta}\big|_{\Theta^{(t)}}\right)\right\| = \rho\left\|\frac{\partial^2 L_i^{meta}(\hat{\mathbf{w}})}{\partial\hat{\mathbf{w}}^2}\big|_{\hat{\mathbf{w}}^{(t)}}\frac{-\alpha}{n}\sum_{j=1}^{n}\frac{\partial L_j^{train}(\mathbf{w})}{\partial\mathbf{w}}\big|_{\mathbf{w}^{(t)}}\frac{\partial v_j(\Theta)}{\partial\Theta}\big|_{\Theta^{(t)}}\right\| \leq \alpha L\rho^3.$

since $\left\|\frac{\partial^2 L_i^{meta}(\hat{\mathbf{w}})}{\partial\hat{\mathbf{w}}^2}\big|_{\hat{\mathbf{w}}^{(t)}}\right\| \leq L, \left\|\frac{\partial L_j^{train}(\mathbf{w})}{\partial\mathbf{w}}\big|_{\mathbf{w}^{(t)}}\right\| \leq \rho, \left\|\frac{\partial v_j(\Theta)}{\partial\Theta}\big|_{\Theta^{(t)}}\right\| \leq \rho$. And for the second term $\left\|(G_{ij})\frac{\partial^2 v_j(\Theta)}{\partial\Theta^2}\big|_{\Theta^{(t)}}\right\| =$

$\left\|\frac{\partial L_i^{meta}(\hat{\mathbf{w}})}{\partial\hat{\mathbf{w}}}\big|_{\hat{\mathbf{w}}^{(t)}}^T\frac{\partial L_j^{train}(\mathbf{w})}{\partial\mathbf{w}}\big|_{\mathbf{w}^{(t)}}\frac{\partial^2 v_j(\Theta)}{\partial\Theta^2}\big|_{\Theta^{(t)}}\right\| \leq \mathcal{B}\rho^2$, since $\left\|\frac{\partial L_i^{meta}(\hat{\mathbf{w}})}{\partial\hat{\mathbf{w}}}\big|_{\hat{\mathbf{w}}^{(t)}}^T\right\| \leq \rho, \left\|\frac{\partial^2 v_j(\Theta)}{\partial\Theta^2}\big|_{\Theta^{(t)}}\right\| \leq \mathcal{B}$. Combining the above two inequalities, we have $\left\|\nabla_{\Theta^2}^2 L_i^{meta}(\hat{\mathbf{w}}^{(t)}(\Theta))\big|_{\Theta^{(t)}}\right\| \leq \alpha(\alpha L\rho^3 + \mathcal{B}\rho^2)$. Therefore, using Lagrange's mean value theorem, Eq. (9) holds.

**Q3.3: The proof of theorem 2 at line 71 in the SM.** We have skipped several steps, and the detailed proof is as follows: Taking expectation of both sides of (21) and since $\mathbb{E}[\psi^{(t)}] = 0$ (line 66-68 of SM), we have

$$\mathbb{E}[\mathcal{L}^{train}(\mathbf{w}^{(t+1)}; \Theta^{(t+1)})] - \mathbb{E}[\mathcal{L}^{train}(\mathbf{w}^{(t)}; \Theta^{(t+1)})] \leq -\alpha_t\mathbb{E}[\|\nabla\mathcal{L}^{train}(\mathbf{w}^{(t)}; \Theta^{(t+1)})\|_2^2] + \frac{L\alpha_t^2}{2}\{\mathbb{E}[\|\nabla\mathcal{L}^{train}(\mathbf{w}^{(t)}; \Theta^{(t+1)})\|_2^2] + \mathbb{E}[\|\psi^{(t)}\|_2^2]\}$$

Summing up the above inequalities over $t = 1, ..., \infty$ in both sides, we obtain (There exists a typo at line 71, $\|\cdot\|$ should be $\|\cdot\|_2^2$)

$$\sum_{t=1}^{\infty}\alpha_t\mathbb{E}[\|\nabla\mathcal{L}^{tr}(\mathbf{w}^{(t)}; \Theta^{(t+1)})\|_2^2] \leq \sum_{t=1}^{\infty}\frac{L\alpha_t^2}{2}\{\mathbb{E}[\|\nabla\mathcal{L}^{tr}(\mathbf{w}^{(t)}; \Theta^{(t+1)})\|_2^2] + \mathbb{E}[\|\psi^{(t)}\|_2^2]\} + \mathbb{E}[\mathcal{L}^{tr}(\mathbf{w}^{(1)}; \Theta^{(2)})] - \lim_{T\to\infty}\mathbb{E}[\mathcal{L}^{tr}(\mathbf{w}^{(t+1)}; \Theta^{(t+1)})] \leq \sum_{t=1}^{\infty}\frac{L\alpha_t^2}{2}\{\rho^2 + \sigma^2\} + \mathbb{E}[\mathcal{L}^{tr}(\mathbf{w}^{(1)}; \Theta^{(2)})] \leq \infty,$$

where $tr$ is short for $train$ to save space, since $\mathbb{E}[\|\nabla\mathcal{L}^{train}(\mathbf{w}^{(t)}; \Theta^{(t+1)})\|_2^2] \leq \rho^2, \mathbb{E}[\|\psi^{(t)}\|_2^2] \leq \sigma^2$.

**Q3.4: Some typos and a different bound for $\mathcal{L}^{(meta)}$ instead of $\rho$.** Yes, there is a $L_2$ norm in the sixth line of Eq. (24), and we'll modify this and other typos in revision. We assume the gradients with respect to training/meta data are both $\rho$-bounded for avoiding the abuse of symbols.

[Meta-Review · NeurIPS 2019]

Reviewers are not entirely satisfied with your response, however, they are leaning to a positive overall opinion. Hence I think your paper can be accepted provided (and I am really trusting on you, as there is no way to obligue you) you commit to address in as much as possible the open issues raised by reviewers. In particular, please be extra-careful with the proofs and theoretical analysis of your paper, this is critical for your reputation and that of NeurIPS.